# CROSS-ENTROPY IS ALL YOU NEED TO INVERT THE DATA GENERATING PROCESS

**Patrik Reizinger**[*1], **Alice Bizeul**[*1,2], **Attila Juhos**[*1],
Julia E. Vogt[2], Randall Balestriero[3], Wieland Brendel[1], David Klindt[4]
`{patrik.reizinger, attila.juhos, wieland.brendel}@tuebingen.mpg.de`
`{alice.bizeul, julia.vogt}@inf.ethz.ch, rbalestr@brown.edu, klindt@cshl.edu`

## ABSTRACT

Supervised learning has become a cornerstone of modern machine learning, yet a comprehensive theory explaining its effectiveness remains elusive. Empirical phenomena, such as neural analogy-making and the linear representation hypothesis, suggest that supervised models can learn interpretable factors of variation in a linear fashion. Recent advances in self-supervised learning, particularly nonlinear Independent Component Analysis, have shown that these methods can recover latent structures by inverting the data generating process. We extend these identifiability results to parametric instance discrimination, then show how insights transfer to the ubiquitous setting of supervised learning with cross-entropy minimization. We prove that even in standard classification tasks, models learn representations of ground-truth factors of variation up to a linear transformation under a certain DGP. We corroborate our theoretical contribution with a series of empirical studies. First, using simulated data matching our theoretical assumptions, we demonstrate successful disentanglement of latent factors. Second, we show that on DisLib, a widely-used disentanglement benchmark, simple classification tasks recover latent structures up to linear transformations. Finally, we reveal that models trained on ImageNet encode representations that permit linear decoding of proxy factors of variation. Together, our theoretical findings and experiments offer a compelling explanation for recent observations of linear representations, such as superposition in neural networks. This work takes a significant step toward a cohesive theory that accounts for the unreasonable effectiveness of supervised learning.

## 1 INTRODUCTION

Representation learning is a central task in machine learning, underpinning the success of extracting and encoding meaningful information from data (Bengio et al., 2013). Among the various paradigms, supervised learning—particularly classification tasks using cross-entropy minimization—has become the dominant method in deep learning (Krizhevsky et al., 2012). Despite its simplicity, this form of supervised learning has led to several intriguing and widely-observed phenomena, including: *neural analogy making* (Mikolov et al., 2013), where models seemingly map between related concepts; the *linear representation hypothesis* (Park et al., 2023), which posits that interpretable features can be linearly decoded from neural representations; recent work on *superposition* in neural networks (Elhage et al., 2022), showing evidence that interpretable features are linearly represented in neural activations (Templeton et al., 2024); and the success of *transfer learning* (Donahue et al., 2014), where a linear readout can be trained on top of learned representations to solve new tasks. These phenomena suggest that deep learning models encode various features in a manner that allows for linear decoding. Yet, a comprehensive theory that explains why these properties emerge in deep learning models has remained elusive (Arora et al., 2016; Park et al., 2023).

We address this gap by building on the theory of Independent Component Analysis (ICA), which studies the conditions under which latent variables in probabilistic models can be uniquely identified (Comon, 1994; Hyvarinen et al., 2001). Recently, ICA has been extended to nonlinear models (Hyvärinen et al., 2023), providing a theoretical foundation for recovering latent variables in a broad class of machine learning tasks (Hyvarinen & Morioka, 2016; Hyvarinen et al., 2019; Gresele et al., 2019; Khemakhem et al., 2020a; Klindt et al., 2021; Khemakhem et al., 2020b; Locatello et al., 2020;

---

[*]Joint first authorship; [1]Max Planck Institute for Intelligent Systems, Tübingen AI Center, ELLIS Institute, Tübingen, Germany; [2]Department of Computer Science, ETH Zürich and ETH AI Center, ETH Zürich, Zürich, Switzerland; [3]Department of Computer Science, Brown University, Rhode Island, USA; [4]Cold Spring Harbor Laboratory, Cold Spring Harbor, New York, USA;

Figure 1: **DIET (Ibrahim et al., 2024) learns identifiable features**: given $N$ samples and a $d-$dimensional latent representation, DIET learns a linear $(N \times d)-$dimensional classification head $\boldsymbol{W}$ on top of a nonlinear encoder $\boldsymbol{f}$ through an instance discrimination objective (1). For unit-normalized $\boldsymbol{f}(\boldsymbol{x}_n)$, DIET maps samples and their augmentations close to the cluster vector $\boldsymbol{v}_c$ corresponding to the class—as if sampled from a von Mises-Fisher (vMF) distribution, centered around $\boldsymbol{v}_c$. For duplicate samples, i.e., matching class labels, the corresponding rows of $\boldsymbol{W}$ will be the same, as shown for $\boldsymbol{x}_1$ and $\boldsymbol{x}_i$ with $\boldsymbol{w}_1 = \boldsymbol{w}_i$.

Morioka et al., 2021; Hälvä et al., 2021; Morioka & Hyvarinen, 2023). Most of these advances have focused on self-supervised learning (SSL) (Hyvarinen & Morioka, 2016; Hyvarinen et al., 2019; Zimmermann et al., 2021; von Kügelgen et al., 2021; Rusak et al., 2024), i.e., when neural networks are trained by solving a surrogate (classification) task to learn from unlabeled data—the exceptions that study supervised learning, though either in the multitask setting, or with a single task with additional assumptions, include (Ahuja et al., 2022; Lachapelle et al., 2023; Fumero et al., 2023). However, we seek to understand whether similar identifiability guarantees can explain under what conditions cross-entropy-based supervised learning, i.e., when the labels for the classification task are provided in the dataset, recovers interpretable and transferable representations.

Our journey starts with a recent development in SSL: nonlinear ICA has been shown to provide identifiability guarantees in contrastive learning, where models invert the data generating process (DGP) and recover latent variables up to linear transformations (Hyvarinen et al., 2019; Zimmermann et al., 2021). Building on this insight, we first extend nonlinear ICA to a simple form of SSL—i.e., parametric instance discrimination (PID) (Dosovitskiy et al., 2014)—through the DIET method (Ibrahim et al., 2024), which streamlines the auxiliary task into an instance-discrimination paradigm. We model the DGP in a new, cluster-centric way, and show that DIET's learned representation is linearly related to the ground-truth representation.

From this foundation, we take the crucial step of extending the theoretical framework to the more common paradigm of supervised learning. Specifically, we show that models can recover ground-truth latent variables up to a linear transformation even in standard classification tasks using the cross-entropy loss, which is the most prevalent setting in modern machine learning. By doing so, we aim to explain why deep learning, particularly supervised classification, is so effective in learning interpretable and transferable representations, offering a unifying framework to explain phenomena such as linear representations and neural analogy-making. Thus, our theoretical insights offer a potential explanation for the extraordinary success of supervised deep learning across a wide variety of tasks. Our **contributions** are

- We propose a cluster-centric DGP as a model for the parametric instance discrimination method of Ibrahim et al. (2024) and prove the DGP's linear identifiability (Thm. 1);
- We use our insight to extend the identifiability guarantee to standard cross-entropy-based supervised classification under the a cluster-centric DGP (Thm. 2);
- We provide a "genealogy" of cross-entropy-based classification methods to connect our identifiability results in instance discrimination and supervised classification to auxiliary-variable nonlinear Independent Component Analysis (ICA) (Hyvarinen et al., 2019) and self-supervised learning (SSL) (§ 3.4) (Zimmermann et al., 2021);
- We corroborate our findings in synthetic experiments matching our cluster-centric DGP, the DisLib disentanglement benchmark (Locatello et al., 2019), and real-world ImageNet-X data (Idrissi et al., 2022), showing that the cross-entropy loss, irrespective of the meaningfulness of labels, can lead to linear identifiability of the features (§ 4).

## 2 BACKGROUND

**Empirical evidence of a linear latent representation.** The *linear representation hypothesis* (Park et al., 2023) has lately received a lot of attention. A weak version of this hypothesis could mean that there are directions in neural activation space that correspond to interpretable features. In the

case of *neural analogy making*, Mikolov et al. (2013) showed that there exist directions in word embeddings that are interpretable and preserved across input pairs. As an example for encoder $\boldsymbol{f}$, producing latent variables $\boldsymbol{z}$, the direction $\boldsymbol{z} = \boldsymbol{f}(man) - \boldsymbol{f}(woman)$ seems to correspond to gender and can be added to other words such as $\boldsymbol{f}(king) + \boldsymbol{z} \approx \boldsymbol{f}(queen)$. Several datasets, such as the Google Analogy Dataset (GA) (Mikolov, 2013) and BATS (Drozd et al., 2016), have been developed to evaluate neural analogy-making. These were, for instance, evaluated in (Dufter & Schütze, 2019). Theoretical explanations of linear representations have been proposed for word embeddings by Arora et al. (2016) and Allen & Hospedales (2019). Both approaches take a statistical learning theory perspective and focus on characterizing the pointwise mutual information. They do not consider cross-entropy-based classification; and, thus, do not make a connection to supervised classification, as we do in Thm. 2. Park et al. (2023) provide a framework to specify what exactly is meant by the linear representation hypothesis. They also provide a strong, causal hypothesis where finding that a feature is linearly represented does not imply that an intervention on that linear subspace will causally remove the feature from the model output. Engels et al. (2024) point out that some latent representations are not linear. This makes intuitive sense if we consider that some latent features, such as the pose of an object, have a non-Euclidean topology that will have to be embedded on a curved manifold in a linear subspace of the latent representation (Higgins et al., 2018; Pfau et al., 2020; Keurti et al., 2023). For instance, the quadrature pair of sines and cosines representing rotations in a $2D$ subspace in (Klindt et al., 2021, Fig. 15) depends on the object symmetries (Bouchacourt et al., 2021). Roeder et al. (2020) prove that different models trained with a discriminative objective converge to learning the same latent representation. Importantly, their claim is about the linear relationship between *any two learned* representations, and not the learned and the ground-truth one, as is usually the case in identifiability theory (Hyvarinen et al., 2001). They also show this empirically for pairs of models trained on different datasets. Their results are corroborated even with widely varying training factors by Moschella et al. (2023). These findings are also supported by recent large scale empirical studies in the converging representations of vision models (Chen & Bonner, 2023). This could also explain the recently proposed *platonic representation hypothesis* (Huh et al., 2024) about the convergence of representations, the improved disentanglement across model families (Du & Xiang, 2021), and the better identifiability of biological mechanisms (Genkin & Engel, 2020). However, these insights from the literature fail to connect the linearity of learned representations to the identifiability of the assumed ground-truth DGP—this is the gap our contribution aims to address.

**Identifiable weakly-/self-supervised learning and ICA.** Independent Component Analysis (ICA) theory studies the conditions under which latent variables in probabilistic models can be uniquely identified (Comon, 1994; Hyvarinen et al., 2001). *Identifiability* means that the learned representation, at the global optimum of the training loss, relates to the *ground-truth* representation (i.e., the ground-truth latent variables underlying the data) via a "simple" transformation, such as permutations or elementwise invertible transformations—this is different to investigations relating two instances of learned representations, such as in Roeder et al. (2020); Moschella et al. (2023); Zhang et al. (2023). Recently, ICA has been extended to nonlinear models (Hyvärinen et al., 2023), providing a theoretical foundation for recovering latent variables in a broad class of learning tasks (Hyvarinen & Morioka, 2016; Hyvarinen et al., 2019; Gresele et al., 2019; Khemakhem et al., 2020a; Klindt et al., 2021; Khemakhem et al., 2020b; Locatello et al., 2020; Morioka et al., 2021; Hälvä et al., 2021; Morioka & Hyvarinen, 2023). Most of these advances have focused on SSL, (Hyvarinen & Morioka, 2016; Hyvarinen et al., 2019; Zimmermann et al., 2021; von Kügelgen et al., 2021; Rusak et al., 2024).

## 3 THEORY

This section presents our main theoretical contribution. We start with our motivation to understand self-supervised learning (SSL) with the help of the simplified DIET (Ibrahim et al., 2024) algorithmic pipeline. For this, we propose a cluster-centric data generating process (DGP) that can model semantic classes (§ 3.1). Then we state our main result in § 3.2 and discuss an intuition behind the identifiability of the representation learned by DIET. We conclude by investigating how DIET fits into the vast literature of (identifiable) SSL and auxiliary-variable Independent Component Analysis (ICA) methods (§ 3.4). This leads to a significant result for proving the identifiability of the latents learned via supervised classification under the DIET DGP (§ 3.3). We provide the technical details for Generalized Contrastive Learning (GCL) (Hyvarinen et al., 2019) in Appx. B.1 and InfoNCE (Chen et al., 2020; Zimmermann et al., 2021) in Appx. B.3.

**Motivation.** Despite significant theoretical progress (Zimmermann et al., 2021; von Kügelgen et al., 2021; Rusak et al., 2024), it remains elusive why SSL methods work well in practice. Rusak et al. (2024) highlighted two remaining gaps between theory and practice: 1) practitioners often discard

the encoder's last few layers (termed the projector) for better performance, despite identifiability guarantees not reflecting this fact; and 2) the data is presumably clustered, not reflected in the common assumption of a uniform marginal. Despite a similar terminology in auxiliary-variable nonlinear ICA algorithms, such as Time-Contrastive Learning (TCL) (Hyvarinen & Morioka, 2016) or GCL (Hyvarinen et al., 2019), it is unclear how such methods relate to SSL at large. Interestingly, the identifiability proofs for nonlinear ICA partition the model into a separate encoder and a regression function (Hyvarinen & Morioka, 2016; Hyvarinen et al., 2019) and prove identifiability for the latent variables after the encoder, but before the regression function. This aligns with the practice of discarding the projector in SSL (Bordes et al., 2023), though identifiability results do not reflect this fact (Zimmermann et al., 2021; von Kügelgen et al., 2021; Rusak et al., 2024). These observations served as our motivation to investigate

*How can we extend the identifiability guarantees to more realistic self-supervised classification scenarios, and can we apply these insights to improve our understanding of supervised learning?*

**Results overview.** We aim to advance our theoretical understanding of SSL, for this, we use the recently proposed DIET (Ibrahim et al., 2024) (detailed in § 3.1), which, beyond its simplicity, promises the strongest and most realistic results, based on similarities to GCL (Hyvarinen et al., 2019). Namely, DIET uses a separate encoder and classification head, and solves an auxiliary classification task akin to GCL—furthermore, its loss correlates with downstream performance, a non-obvious and welcome fact (Rusak et al., 2024). This provides the hope to resolve the two above points by modeling the cluster structure of the data and proving identifiability for the representation used for downstream tasks (Thm. 1). Subsequently, we leverage the insights from our identifiability theory and the DIET pipeline's similarity to *supervised* classification to show how the latter is a special case of DIET, where the sample indices correspond to the semantic class labels (Thm. 2).

### 3.1 SETUP

**DIET (Ibrahim et al., 2024).** DIET solves an instance classification problem, where each sample $x$ in the training dataset of size $N$ has a unique instance label $i$. Augmentations do not affect this label. We have a composite model $W \circ f$, where the backbone $f$ produces $d$-dimensional representations, and a linear, bias-free classification head $W \in \mathbb{R}^{N \times d}$ maps these representations to a logit vector equal in size to the cardinality of the training dataset. If the parameter vector corresponding to logit $i$ is denoted as $w_i$, then $W$ effectively computes similarity scores (scalar products) between the $w_i$'s and embeddings $f(x)$. DIET trains this architecture to predict the correct instance label using multinomial regression (with $f, W$ and temperature $\beta$ as learnable variables), i.e., it solves a parametric instance discrimination (PID) task (Dosovitskiy et al., 2014; Wu et al., 2018):

$$\mathcal{L}_{\text{PID}}(f, W, \beta) = \mathbb{E}_{(x,i)} \left[ -\ln \frac{e^{\beta \langle w_i, f(x) \rangle}}{\sum_j e^{\beta \langle w_j, f(x) \rangle}} \right]. \tag{1}$$

An important fact is that (1) is the cross-entropy loss with instance labels, which we will leverage to connect instance discrimination to supervised classification.

**The proposed cluster-centric data generating process (DGP).** To prove the identifiability of the latent variables, we need to formally define a latent variable model (LVM) for the data generating process (DGP). We take a cluster-centric approach, representing semantic classes by cluster vectors, similar to proxy-based metric learning (Kirchhof et al., 2022). Then, we model the samples of a class with a von Mises-Fisher (vMF) distribution (intuitively, this is an isotropic multivariate Normal distribution that is restricted to the unit hypershere), centered around the class's cluster vector. This conditional distribution jointly models intra-class sample selection and *augmentations* of samples, together called *intra-class variances*. In contrast to conventional SSL methods such as InfoNCE (Zimmermann et al., 2021), this conceptually separates global and local structure in the latent space: 1) the cluster-vectors describe the global structure of the latent space; and 2) the cluster-centric conditional in (2) describes the local structure. This cluster-centric conditional embodies that data augmentations are selected such that they ought not to change the sample's semantic class. Our conditional does not mean that each sample pair transforms into each other via augmentations *with high probability*. It does mean that—since we assume a latent variable model (LVM) on the hypersphere; i.e., all semantic concepts (color, position, etc.) correspond to a continuous latent variable—the latent manifold is connected, or equivalently, that the augmentation graph is connected, which is an assumption used in (Wang et al., 2022; Balestriero & LeCun, 2022; HaoChen et al., 2022). We provide an overview of our assumptions, and defer additional details to Assums. 1C in Appx. A:

**Assumptions 1** (DGP with vMF samples around cluster vectors. *Simplified.*)**.**

*(i) There is a finite set of semantic classes $\mathscr{C}$, represented by a set of unit-norm $d$-dimensional cluster-vectors $\{\boldsymbol{v}_c | c \in \mathscr{C}\} \subseteq \mathbb{S}^{d-1}$. The system $\{\boldsymbol{v}_c\}$ is sufficiently large and spread out.*

*(ii) Any instance label $i$ belongs to exactly one class $c = \mathcal{C}(i)$.*

*(iii) The latent variable $\boldsymbol{z} \in \mathbb{S}^{d-1}$ of our data sample with instance label $i$ is drawn from a vMF distribution with concentration parameter $\kappa$ around the cluster vector $\boldsymbol{v}_c$ of class $c = \mathcal{C}(i)$:*

$$\boldsymbol{z} \sim p(\boldsymbol{z}|c) \propto e^{\kappa \langle \boldsymbol{v}_c, \boldsymbol{z} \rangle}. \tag{2}$$

*(iv) Sample $\boldsymbol{x}$ is generated by passing latent $\boldsymbol{z}$ through an injective generator function: $\boldsymbol{x} = \boldsymbol{g}(\boldsymbol{z})$.*

## 3.2 MAIN RESULT: DIET IDENTIFIES BOTH LATENT VARIABLES AND CLUSTER VECTORS

Under Assums. 1, we prove the identifiability of both the latent representations $\boldsymbol{z}$ and the cluster vectors, $\boldsymbol{v}_c$, in all four combinations of unit-normalized (i.e., when the latent space is the hypersphere, commonly used, e.g., in InfoNCE (Chen et al., 2020)); and non-normalized (as in the original DIET paper (Ibrahim et al., 2024)) learned embeddings, $\tilde{\boldsymbol{z}}$, and weight vectors, $\boldsymbol{w}_i$. We state a concise version of our result and defer the full treatment and the proof to Thm. 1C in Appx. A:

**Theorem 1** (Identifiability of latent variables drawn from vMF around cluster vectors. *Simplified.*)**.** *Let $(\boldsymbol{f}, \boldsymbol{W}, \beta)$ globally minimize the DIET objective (1) under the following additional constraints:*

*C3. the embeddings $\boldsymbol{f}(\boldsymbol{x})$ are unnormalized, while the $\boldsymbol{w}_i$'s are unit-normalized. Then $\boldsymbol{w}_i$ identifies the cluster vector $\boldsymbol{v}_{\mathcal{C}(i)}$ up to an orthogonal linear transformation $\mathcal{O}$: $\boldsymbol{w}_i = \mathcal{O}\boldsymbol{v}_{\mathcal{C}(i)}$, for any $i$. Furthermore, the inferred latent variables $\tilde{\boldsymbol{z}} = \boldsymbol{f}(\boldsymbol{x})$ identify the ground-truth latent variables $\boldsymbol{z}$ up to a scaled orthogonal transformation with the same $\mathcal{O}$: $\boldsymbol{z} = \frac{\kappa}{\beta}\mathcal{O}\tilde{\boldsymbol{z}}$.*

*C4. neither the embeddings $\boldsymbol{f}(\boldsymbol{x})$ nor the $\boldsymbol{w}_i$'s are unit-normalized. Then $\boldsymbol{w}_i$ identifies the cluster vectors $\boldsymbol{v}_c$ up to an affine linear transformation. Furthermore, the inferred latent variables $\tilde{\boldsymbol{z}}$ identify the ground-truth latent variables $\boldsymbol{z}$ up to a linear transformation.*

*In all cases, the weight vectors belonging to samples of the same class are equal, i.e., for any $i, j$, $\mathcal{C}(i) = \mathcal{C}(j)$ implies $\boldsymbol{w}_i = \boldsymbol{w}_j$.*

**Intuition.** DIET assigns a different (instance) label and a unique weight vector $\boldsymbol{w}_i$ to each training sample. The cross-entropy objective is optimized if the trained neural network can distinguish between the samples. Thus, the learned representation $\tilde{\boldsymbol{z}} = \boldsymbol{f}(\boldsymbol{x})$ should capture enough information to distinguish different samples, even from the same class. However, the weight vectors $\boldsymbol{w}_i$'s cannot be sensitive to the intra-class sample variance or the sample's instance label $i$ (because the conditional distribution over latent variables is identical for all samples of the same class). This leads to the weight vectors taking the values of the cluster vectors. As cluster vectors only capture some statistics of the conditional (1), feature recovery is more fine-grained than cluster identifiability. The interaction between the two is dictated by the cross-entropy loss, which is minimized if the representation $\tilde{\boldsymbol{z}}$ is most similar to its own assigned weight vector $\boldsymbol{w}_i$. Fig. 1 provides a visualization conveying the intuition behind Thm. 1.

## 3.3 SUPERVISED CLASSIFICATION

This section relates our cluster-centric DGP to *supervised* classification. To see how supervised machine learning is a special case of self-supervised approaches, consider that the sample index (i.e., the target of the cross-entropy loss) can be defined *arbitrarily* (as long as Assums. 1 are still satisfied). This means that many labelings are possible, including the one used for supervised classification. This, *in hindsight* obvious insight has important consequences: it can explain the success of supervised cross-entropy-based classification. Namely, supervised learning performs non-linear ICA under our proposed DGP (Assums. 1). We demonstrate this in §§ 4.1 and 4.3. We state a concise version of our result and defer the full treatment to Appx. A:

**Theorem 2** (Identifiability of latent variables drawn from a vMF around class vectors)**.** *Let Assum. 3 hold, and suppose that a continuous encoder $\boldsymbol{f} : \mathbb{R}^D \to \mathbb{R}^d$, a linear classifier $\boldsymbol{W}$ with rows $\{\boldsymbol{w}_c^\top \mid c \in \mathscr{C}\}$, and $\beta > 0$ globally minimize the cross-entropy objective:*

$$\mathcal{L}_{\text{supervised}}(\boldsymbol{f}, \boldsymbol{W}, \beta) = \mathbb{E}_{(\boldsymbol{x}, C)} \left[ -\ln \frac{e^{\beta \langle \boldsymbol{w}_C, \boldsymbol{f}(\boldsymbol{x}) \rangle}}{\sum_{c' \in \mathscr{C}} e^{\beta \langle \boldsymbol{w}_{c'}, \boldsymbol{f}(\boldsymbol{x}) \rangle}} \right].$$

*Then, the composition $\boldsymbol{h} = \boldsymbol{f} \circ \boldsymbol{g}$ is a linear map from $\mathbb{S}^{d-1}$ to $\mathbb{R}^d$.*

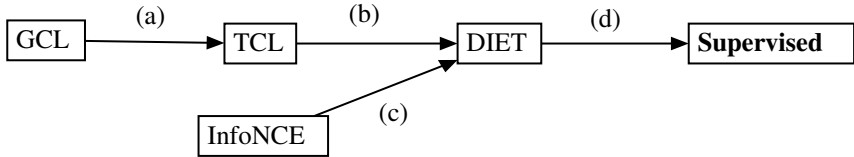

Figure 2: **The simplified genealogy of cross-entropy-based classification methods** (cf. Tab. 1 for details): The labeled arrows express how to go from general to special methods. **(a)** The most general auxiliary-variable ICA framework, Generalized Contrastive Learning (GCL) (Hyvarinen et al., 2019), yields Time-Contrastive Learning (TCL) (Hyvarinen & Morioka, 2016) as the special case when the latent conditional is assumed to come from an exponential family (of order one) with a scalar auxiliary variable; **(b)** TCL relates to non-unit-normalized DIET by further restricting the latent conditional to a vMF distribution; **(c)** if the neural network used in InfoNCE is partitioned into a linear classifier head and a backbone, the marginal is assumed to be a vMF instead of uniform, we get the unit-normalized version of DIET; **(d)** if the labeling function in DIET is assumed to assign the semantic class labels to the samples, we get classic supervised training

**Intuition:** In the context of DIET, the cross-entropy objective encourages the learned representations to align with the cluster vectors corresponding to each class. The identifiability of the latent variables is ensured by the fact that the cluster structure reflects the underlying data distribution, modeled as a vMF distribution. This leads to a representation that captures the latent structure up to an orthogonal transformation. *Given the same underlying structure as in DIET, supervised learning can be viewed as a special case of instance discrimination, where the instance labels are replaced by class labels.* The cross-entropy objective, when applied to classification tasks and assuming our DGP from Assums. 1, similarly encourages representations to align with class vectors. As a result, the latent variables are recovered up to a linear transformation, providing a theoretical explanation for the success of supervised classification in learning linearly decodable representations.

### 3.4 THE GENEALOGY OF IDENTIFIABLE CLASSIFICATION WITH CROSS-ENTROPY

Our main result in Thm. 1, and its corollary for supervised classification (Thm. 2) suggest the following surprising conclusion to invert the proposed DGP (Assums. 1):

*Solving an (almost) arbitrary classification task by optimizing the cross-entropy objective is sufficient to invert the DGP and identify the ground-truth representation up to a linear transformation.*

To show how solving a cross-entropy-based classification task is a key component to invert the DGP and to achieve linear identifiability, we provide a unified treatment of auxiliary-variable ICA (i.e., weakly supervised or self-supervised classification) and supervised classification methods. We call this a *genealogy* to allude to the fact that these methods can be seen as special cases, descending from each other (cf. Fig. 2 and Tab. 1 for an overview, and Appx. B for details).

**From GCL to TCL (Fig. 2a: arbitrary scalar labels and exponential family latent variables).** The most general framework we consider is Generalized Contrastive Learning (GCL) (Hyvarinen et al., 2019), i.e., auxiliary-variable nonlinear ICA. GCL works with conditionally independent latent variables in Euclidean space given (possibly vector-valued) auxiliary information $\mathbf{u}$. It aims to classify different values of $\mathbf{u}$ by distinguishing $(\boldsymbol{x}, \mathbf{u})$ from $(\boldsymbol{x}, \mathbf{u}^*)$, where $\mathbf{u}^*$ is an arbitrary value of the auxiliary variable. At the Bayes optimum of the cross-entropy loss, GCL provides identifiability of the latent variables after the encoder $\boldsymbol{f}$, but before the classifier head $\boldsymbol{W}$, up to elementwise invertible transformations. When the latent variables are distributed

Table 1: **Comparison of the components of different cross-entropy-based classification methods:** $\mathbf{u}$ denotes a (possibly) vector-valued auxiliary variable, $t$ is the scalar time step, $i$ the sample index, and $c$ the semantic class; ExpFam stands for exponential family, $\perp^{\mathbf{u}}$ for conditionally independent sources given the auxiliary variable, $\boldsymbol{W}$ is the classifier head, $\boldsymbol{f}$ the encoder, whereas N/A stands for no assumption

| Property | GCL | TCL | InfoNCE | DIET | Supervised |
|---|---|---|---|---|---|
| Latent space | $\mathbb{R}^d$ | $\mathbb{R}^d$ | $\mathcal{S}^{d-1}$ | $\mathbb{R}^d/\mathcal{S}^{d-1}$ | $\mathbb{R}^d$ |
| Network | $\boldsymbol{W} \circ \boldsymbol{f}$ | $\boldsymbol{W} \circ \boldsymbol{f}$ | $\boldsymbol{f}$ | $\boldsymbol{W} \circ \boldsymbol{f}$ | $\boldsymbol{W} \circ \boldsymbol{f}$ |
| Aux.info | $\mathbf{u}$ | $t$ | $i$ | $i$ | $c$ |
| Conditional | $\perp^{\mathbf{u}}$ | ExpFam | vMF | vMF | vMF |
| Marginal | N/A | N/A | uniform | uniform | uniform |

Table 2: **Identifiability results for parametric instance discrimination (PID) in numerical simulations:** Mean $\pm$ standard deviation across 5 random seeds. Settings that match and violate our theoretical assumptions are denoted as ✓ and ✗, respectively. We report the $R^2$ score for linear maps $\tilde{z} \to z$ and $w_i \to v_c$ with normalized (subscript $o$) and not normalized (subscript $a$) $w_i$. For normalized $w_i$, we verify that the $\tilde{z} \to z$ maps are orthogonal by reporting the Mean Absolute Error (MAE) between their singular values and those of an orthogonal transformation.

| | | | | | normalized $w_i$ | | | | unnormalized $w_i$ | |
| | | | | | $R_o^2(\uparrow)$ | | $\mathrm{MAE_o}(\downarrow)$ | | $R_a^2(\uparrow)$ | |
| $N$ | $d$ | $|\mathscr{C}|$ | $p(z\|v_c)$ | M. | $\tilde{z} \to z$ | $w_i \to v_c$ | $\tilde{z} \to z$ | $w_i \to v_c$ | $\tilde{z} \to z$ | $w_i \to v_c$ |
|---|---|---|---|---|---|---|---|---|---|---|
| $10^3$ | 5 | 100 | vMF($\kappa{=}10$) | ✓ | $98.6_{\pm 0.01}$ | $99.9_{\pm 0.00}$ | $0.01_{\pm 0.00}$ | $0.00_{\pm 0.00}$ | $99.0_{\pm 0.00}$ | $99.9_{\pm 0.00}$ |
| $10^5$ | 5 | 100 | vMF($\kappa{=}10$) | ✓ | $98.2_{\pm 0.01}$ | $99.5_{\pm 0.00}$ | $0.00_{\pm 0.00}$ | $0.00_{\pm 0.00}$ | $99.7_{\pm 0.00}$ | $99.8_{\pm 0.00}$ |
| $10^3$ | 5 | 100 | vMF($\kappa{=}10$) | ✓ | $98.6_{\pm 0.01}$ | $99.9_{\pm 0.00}$ | $0.01_{\pm 0.00}$ | $0.00_{\pm 0.00}$ | $99.0_{\pm 0.00}$ | $99.9_{\pm 0.00}$ |
| $10^3$ | 10 | 100 | vMF($\kappa{=}10$) | ✓ | $92.5_{\pm 0.01}$ | $99.6_{\pm 0.00}$ | $0.01_{\pm 0.00}$ | $0.00_{\pm 0.00}$ | $93.0_{\pm 0.03}$ | $99.6_{\pm 0.00}$ |
| $10^3$ | 20 | 100 | vMF($\kappa{=}10$) | ✓ | $70.8_{\pm 0.02}$ | $97.1_{\pm 0.01}$ | $0.03_{\pm 0.00}$ | $0.00_{\pm 0.00}$ | $81.9_{\pm 0.01}$ | $99.7_{\pm 0.00}$ |
| $10^3$ | 5 | 10 | vMF($\kappa{=}10$) | ✓ | $88.6_{\pm 0.05}$ | $85.7_{\pm 0.15}$ | $0.02_{\pm 0.00}$ | $0.00_{\pm 0.00}$ | $90.0_{\pm 0.05}$ | $99.0_{\pm 0.03}$ |
| $10^3$ | 5 | 100 | vMF($\kappa{=}10$) | ✓ | $98.6_{\pm 0.01}$ | $99.9_{\pm 0.01}$ | $0.01_{\pm 0.00}$ | $0.00_{\pm 0.00}$ | $99.0_{\pm 0.00}$ | $99.9_{\pm 0.00}$ |
| $10^3$ | 5 | 1000 | vMF($\kappa{=}10$) | ✓ | $99.3_{\pm 0.00}$ | $99.9_{\pm 0.00}$ | $0.00_{\pm 0.00}$ | $0.00_{\pm 0.00}$ | $99.2_{\pm 0.00}$ | $99.9_{\pm 0.00}$ |
| $10^3$ | 5 | 100 | vMF($\kappa{=}5$) | ✓ | $98.6_{\pm 0.01}$ | $99.9_{\pm 0.01}$ | $0.01_{\pm 0.00}$ | $0.00_{\pm 0.00}$ | $99.0_{\pm 0.00}$ | $99.8_{\pm 0.00}$ |
| $10^3$ | 5 | 100 | vMF($\kappa{=}10$) | ✓ | $99.0_{\pm 0.00}$ | $99.9_{\pm 0.00}$ | $0.00_{\pm 0.00}$ | $0.00_{\pm 0.00}$ | $99.1_{\pm 0.00}$ | $99.9_{\pm 0.00}$ |
| $10^3$ | 5 | 100 | vMF($\kappa{=}50$) | ✓ | $45.0_{\pm 0.06}$ | $49.7_{\pm 0.06}$ | $0.30_{\pm 0.00}$ | $0.00_{\pm 0.00}$ | $72.5_{\pm 0.03}$ | $75.5_{\pm 0.00}$ |
| $10^3$ | 5 | 100 | vMF($\kappa{=}10$) | ✓ | $98.6_{\pm 0.01}$ | $99.9_{\pm 0.01}$ | $0.01_{\pm 0.00}$ | $0.00_{\pm 0.00}$ | $99.0_{\pm 0.00}$ | $99.9_{\pm 0.00}$ |
| $10^3$ | 5 | 100 | Laplace ($b{=}1.0$) | ✗ | $85.2_{\pm 0.01}$ | $99.7_{\pm 0.01}$ | $0.01_{\pm 0.00}$ | $0.00_{\pm 0.00}$ | $85.4_{\pm 0.00}$ | $99.5_{\pm 0.00}$ |
| $10^3$ | 5 | 100 | Normal ($\sigma^2{=}1.0$) | ✗ | $98.7_{\pm 0.00}$ | $99.8_{\pm 0.00}$ | $0.01_{\pm 0.00}$ | $0.00_{\pm 0.00}$ | $98.6_{\pm 0.00}$ | $99.6_{\pm 0.00}$ |

according to an exponential family distribution and the auxiliary variable is a scalar (e.g., time), then we get the more specialized method, named Time-Contrastive Learning (TCL) (Hyvarinen & Morioka, 2016). If the order of the exponential family is one, identifiability holds only up to a linear transformation, otherwise, up to elementwise invertible transformations.

**From TCL to DIET (Fig. 2b: sample index as $u$ and vMF latent variables).** Using our cluster-centric DGP (Assums. 1), and assuming an even more special latent distribution (i.e., a vMF), we get the identifiability guarantee for DIET, i.e., our main result in Thm. 1. The auxiliary variable is a scalar for our result, too; however, instead of time, it is the (arbitrary) sample index.

**From InfoNCE to DIET (Fig. 2c: a compositional model $W \circ f$ and unit-normalized latent variables).** Importantly, our main result also encompasses unit-normalized representations, the conventional choice in (identifiable) SSL such as InfoNCE (cf. Appx. B.3 for details on InfoNCE)—this is why we illustrate both InfoNCE and TCL as being the "parents" of DIET in Fig. 2. Thus, Thm. 1 is more general in terms of latent spaces than nonlinear ICA, and it proves identifiability for the latent variables that are used post-training, as opposed to the proofs for InfoNCE in (Zimmermann et al., 2021; Rusak et al., 2024), where practitioners discard the last few layers.

**From DIET to supervised classification (Fig. 2d: semantic class labels).** When the labeling function assigns the semantic class labels, and not arbitrary indices, then our identifiability result still holds, yielding the case of supervised learning (Thm. 2).

## 4 EMPIRICAL RESULTS

In § 4.1, we empirically verify the claims made in Thm. 1 and Thm. 2 in the synthetic setting. We generate data samples according to Assums. 1: ground-truth latent variables are sampled around cluster centroids $v_c$ following a vMF distribution. Data augmentations, which share the same instance label $i$, are sampled from the same vMF distribution around $v_c$. In § 4.2, we describe our results on the DisLib disentanglement benchmark (Locatello et al., 2019), and § 4.3 includes our experiments on ImageNet-X (Idrissi et al., 2022). We made our code publicly available on GitHub[1].

---

[1]https://github.com/klindtlab/csi

## 4.1 SYNTHETIC DATA

**Setup.** We consider $N$ latent samples of dimensionality $d$ generated from the conditional vMF $z \sim p(z|v_c)$, sampled around a set of $|\mathscr{C}|$ class vectors $v_c$, which are uniformly distributed across the unit hyper-sphere $\mathcal{S}^{d-1}$. We use an invertible multi-layer perceptron (MLP) to map ground-truth latent variables to data samples. We train a classification head $W = [w_i^\top |_{i=1}^N]$ and an MLP encoder that maps samples to representations $\tilde{z} \in \mathbb{R}^d$ using the DIET objective (1). While to verify Thm. 1 case C4., we do not normalize $W$, we do unit-normalize the weight vectors to validate Thm. 1 case C3. We verify our theoretical claims by measuring the predictability of the ground-truth $z$ from $\tilde{z}$ and $v_c$ from $w_i$ using the $R^2$ score on a held-out dataset (Wright, 1921). For identifiability up to orthogonal linear transformations, we train linear mappings with no intercept, assess the $R^2$ score and verify that the singular values of this transformation converge to 1, while for identifiability up to affine linear transformations, we simply assess the $R^2$ of a linear predictor with intercept.

**Results for DIET.** In Tab. 2, we report the $R^2$ scores for the recovery of the cluster vectors $v_c$ from $W$'s rows and of the ground-truth latent variables $z$ from the learned latent variables $\tilde{z}$. For DIET's PID task, we also consider cases with row-normalized $W$. We observe scores close to $100\%$ ($\geq 98\%$), even with many clusters ($\geq 10^3$) and samples ($\sim 10^5$). High latent dimensionality ($> 10$) does impact the recovery of ground-truth latent variables—such scalability problems are a common artifact in SSL (Zimmermann et al., 2021; Rusak et al., 2024). For a higher concentration of samples around $v_c$ (i.e., $\kappa = 50$) as well as a lower number of clusters (i.e., $|\mathscr{C}| = 10$), the $R^2$ score decreases, which is also a common phenomenon, and is possibly explained by too strong augmentation overlap (Wang et al., 2022; Rusak et al., 2024). For a low number of clusters, high $\kappa$ and a fixed number of training samples, the concentration of samples in regions surrounding centroids, $v_c$, increases, a setting, refered to as "overly overlapping augmentations", known to be suboptimal and leading to a drop in downstream performance (Wang et al., 2022). Our results also suggest that even under model misspecification

| $d$ | $|\mathscr{C}|$ | $p(z|v_c)$ | M. | $R^2 : \tilde{z} \to z$ |
|---|---|---|---|---|
| 5 | 100 | vMF($\kappa = 10$) | ✓ | $99.8_{\pm 0.00}$ |
| 10 | 100 | vMF($\kappa = 10$) | ✓ | $97.2_{\pm 0.01}$ |
| 20 | 100 | vMF($\kappa = 10$) | ✓ | $82.1_{\pm 0.02}$ |
| 5 | 10 | vMF($\kappa = 10$) | ✓ | $97.5_{\pm 0.03}$ |
| 5 | 100 | vMF($\kappa = 10$) | ✓ | $99.8_{\pm 0.00}$ |
| 5 | 1000 | vMF($\kappa = 10$) | ✓ | $99.8_{\pm 0.00}$ |
| 5 | 10000 | vMF($\kappa = 10$) | ✓ | $99.8_{\pm 0.00}$ |
| 5 | 100 | vMF($\kappa = 5$) | ✓ | $99.7_{\pm 0.00}$ |
| 5 | 100 | vMF($\kappa = 10$) | ✓ | $99.7_{\pm 0.00}$ |
| 5 | 100 | vMF($\kappa = 50$) | ✓ | $65.5_{\pm 0.09}$ |
| 5 | 100 | vMF($\kappa = 10$) | ✓ | $99.8_{\pm 0.00}$ |
| 5 | 100 | Laplace ($b = 1.0$) | ✗ | $85.4_{\pm 0.01}$ |
| 5 | 100 | Normal ($\sigma^2 = 1.0$) | ✗ | $99.6_{\pm 0.00}$ |

Table 3: **Identifiability results for supervised learning in numerical simulations:** Mean $\pm$ standard deviation across 5 random seeds. Settings that match and violate our theoretical assumptions are denoted as ✓ and ✗, respectively. We report the $R^2$ score for linear mappings $\tilde{z} \to z$, and not normalized $w_i$. We used $N = 10^3$ samples

(last two rows in Tab. 2 with non-vMF distributions), identifiability still holds. For unit-normalized $W$ rows, the MAE is lower, confirming the orthogonality of the map $w_i \to v_c$. We additionally ablate over batch size, concentration, and conditional in Appx. D.

**Results for Supervised Classification.** In Tab. 3, where the semantic class labels were used instead of the sample index, we only report the $R^2$ score for the recovery of the ground-truth latent variables $z$ from the learned latent variables $\tilde{z}$. In all but one setting, we observe higher $R^2$ from representations learned with class labels rather than instance indices. This suggests that even a coarser classification task may suffice to learn linearly identifiable representations of the underlying latent variables.

## 4.2 DISLIB

**Setup.** Next, we evaluate our methods on the DisLib disentanglement benchmark (Locatello et al., 2019), which provides a controlled setting for testing disentanglement and latent variable recovery. It includes the vision datasets dSprites, Shapes 3D, MPI 3D, Cars 3D, and smallNORB. We train both a three-layer MLP with 512 latent dimensions and BatchNorm (which helped with trainability) and a CNN (ResNet18) also with 512 latent dimensions. We only consider latent variables with Euclidean topology, as non-Euclidean, e.g., periodic latent variables such as orientation, are problematic to learn and are potentially mapped to a nonlinear manifold (Higgins et al., 2018; Pfau et al., 2020; Keurti et al., 2023; Engels et al., 2024). We evaluate the recovery of latent variables by computing the Pearson correlation between ground-truth and predicted factors. We detail our setup in Appx. C.2.

Table 4: **Identifiability in DisLib datasets (Locatello et al., 2019):** We train different models to predict the categorical variable in each setting: $(\boldsymbol{x})$: as a baseline, from the inputs; $(\boldsymbol{f}_{\textbf{MLP}}(\boldsymbol{x}))$: from a three-layer MLP; and $(\boldsymbol{f}_{\textbf{CNN}}(\boldsymbol{x}))$: from a CNN (ResNet18). All continuous latent variables can be decoded from the learned representations, corroborated by the Pearson correlation—reported with mean $\pm$ standard deviation across 3 random seeds. Including the category is informative to see how well the underlying training classification task was solved.

| Model | Latent | $\boldsymbol{x}$ | $\boldsymbol{f}_{\textbf{MLP}}(\boldsymbol{x})$ | $\boldsymbol{f}_{\textbf{CNN}}(\boldsymbol{x})$ |
|---|---|---|---|---|
| dSprites | category | $0.26_{\pm 0.00}$ | $0.94_{\pm 0.01}$ | $\mathbf{1.00}_{\pm 0.00}$ |
| dSprites | scale | $0.62_{\pm 0.00}$ | $\mathbf{0.98}_{\pm 0.00}$ | $0.92_{\pm 0.05}$ |
| dSprites | posX | $0.92_{\pm 0.00}$ | $0.97_{\pm 0.00}$ | $\mathbf{0.99}_{\pm 0.00}$ |
| dSprites | posY | $0.92_{\pm 0.00}$ | $0.97_{\pm 0.00}$ | $\mathbf{0.99}_{\pm 0.00}$ |
| Shapes 3D | category | $0.42_{\pm 0.00}$ | $1.00_{\pm 0.00}$ | $\mathbf{1.00}_{\pm 0.00}$ |
| Shapes 3D | objSize | $0.21_{\pm 0.00}$ | $0.89_{\pm 0.01}$ | $\mathbf{0.99}_{\pm 0.00}$ |
| Shapes 3D | objAzimuth | $0.04_{\pm 0.00}$ | $0.85_{\pm 0.02}$ | $\mathbf{0.93}_{\pm 0.01}$ |
| MPI 3D | category | $0.03_{\pm 0.00}$ | $0.71_{\pm 0.01}$ | $\mathbf{0.97}_{\pm 0.00}$ |
| MPI 3D | posX | $0.28_{\pm 0.00}$ | $0.76_{\pm 0.01}$ | $\mathbf{0.90}_{\pm 0.01}$ |
| MPI 3D | posY | $0.46_{\pm 0.00}$ | $0.76_{\pm 0.01}$ | $\mathbf{0.84}_{\pm 0.01}$ |
| MPI 3D real | category | $0.19_{\pm 0.00}$ | $0.88_{\pm 0.01}$ | $\mathbf{0.98}_{\pm 0.00}$ |
| MPI 3D real | posX | $0.14_{\pm 0.00}$ | $0.74_{\pm 0.01}$ | $\mathbf{0.83}_{\pm 0.01}$ |
| MPI 3D real | posY | $0.44_{\pm 0.00}$ | $0.54_{\pm 0.01}$ | $\mathbf{0.71}_{\pm 0.02}$ |
| Cars 3D | category | $0.05_{\pm 0.00}$ | $0.63_{\pm 0.11}$ | $\mathbf{0.77}_{\pm 0.02}$ |
| Cars 3D | elevation | $0.15_{\pm 0.00}$ | $\mathbf{0.87}_{\pm 0.03}$ | $0.78_{\pm 0.02}$ |
| smallNORB | category | $0.22_{\pm 0.00}$ | $0.94_{\pm 0.01}$ | $\mathbf{1.00}_{\pm 0.00}$ |
| smallNORB | elevation | $0.15_{\pm 0.00}$ | $\mathbf{0.83}_{\pm 0.01}$ | $0.79_{\pm 0.01}$ |

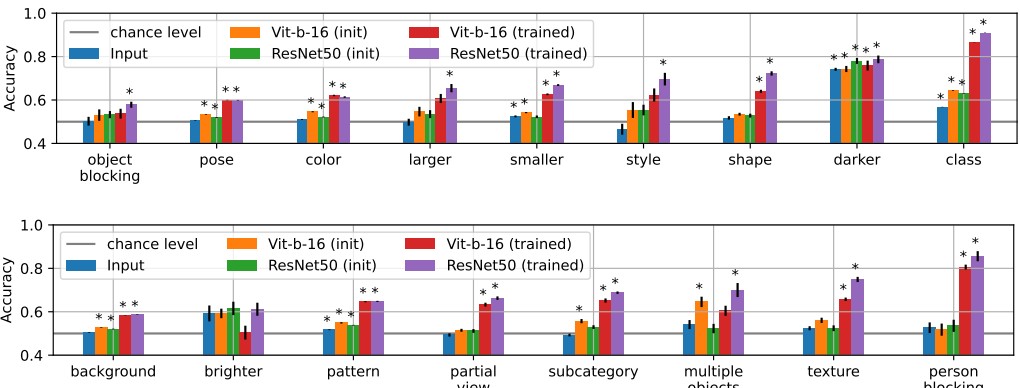

Figure 3: **Approximate identifiability on ImageNet-X against a random (shuffled) baseline:** Using ImageNet-X (Idrissi et al., 2022), we test how well linear decoders are able to predict each latent from the second-to-last layer of different models, i.e., when the classification head is discarded. We train a linear classifier on the features, and plot the accuracy of predicting different latent variables. As baselines, we also try decoding from the raw input and from the randomly initialized model representations. Error-bars indicate standard error of the mean (SEM) across 10 seeds of balanced resampling. Asterisks indicate significant $p$-values (against a null hypothesis of $0.5$ chance level accuracy) at an $\kappa = 0.05/85$ multiple comparison (Bonferroni) adjusted significance level.

**Results.** The models trained using cross-entropy were able to recover latent variables such as object position, scale, and orientation with high accuracy. As shown in Tab. 4, the Pearson correlation is generally highest when predicting the latent variables from the CNN's representation, which we attribute to the CNN's suitable inductive bias for images. In few cases, such as the position in dSprites, this can be done with fairly high accuracy even on the input data. Nevertheless, in all settings the nonlinear function estimated by the model is necessary to linearly identify the correct latent variables.

### 4.3  REAL DATA: IMAGENET-X

**Setup.**  Finally, we test the generalizability of our theoretical insights on real-world data using ImageNet-X (Idrissi et al., 2022). The latent variables are binary proxies, defined by human annotators (Idrissi et al., 2022). We evaluate how well linear decoders can predict latent variables from pretrained model representations. We use two architectures, a ResNet50 and a Vit-b-16 both trained on standard supervised classification using a cross-entropy loss on the full ImageNet dataset (Deng et al., 2009). As baselines, we also decode from the inputs and the randomly initialized models. After balanced sub-sampling, over 10 random seeds, we report accuracies. We use $t$-tests against a chance level of $50\%$ with a Bonferroni adjusted significance level of $\kappa = \frac{0.05}{17 \cdot 5}$. Detail are in Appx. C.3.

**Results.**  Fig. 3 shows that even in complex, high-dimensional data, latents can be linearly decoded from representations learned via supervised learning, in most cases significantly above chance level. Some factors (e.g., *darker* and *brighter*) are linearly decodable even from untrained models or input space. Unsurprisingly, decoding *class* (binarized ImageNet labels, every index $< 500$ is set to $0$ and every index $\geq 500$ is set to $1$) works well for the trained models. ResNet50 has slightly higher decoding performance, possibly due to the larger latent space ($d = 2048$, compared to $d = 768$ in ViT). While texture information may be expected (Geirhos et al., 2018), the presence of shape information suggests that shortcut learning may be mitigated even after standard training (Geirhos et al., 2020).

## 5  DISCUSSION

**Limitations.**  One limitation of our work is that we mainly focus on synthetic and controlled datasets. While the results on ImageNet-X (Idrissi et al., 2022) are promising, they only provide some supporting evidence for our theory on real data. The factors in ImageNet-X are likely not the true latent variables of the data generating process, still, the linear identifiablity results on these proxy latent variables support our theoretical results. Further experiments on other large-scale datasets would support the generality of our findings. However, this would require the availability of such datasets with full latent variable annotations. Although our cluster-centric modeling of the data generating process allows capturing the inherent structure of the data, our assumption about the latent variables' geometric properties (such as being drawn from a vMF distribution on a hypersphere), may not hold in all real-world settings. For instance, the pose of an object in a scene is, arguably, an independent component/subspace corresponding to a point on $SO(3)$, which has a distinct topology from our assumed latent variables on a hypersphere. Moreover, the assumption that a data sample and its augmented version are conditionally independent given their semantic class could be relaxed in future work, since it may be misaligned with realistic scenarios (Wang et al., 2022). Despite these simplifications, our experimental results also suggest that our assumptions can be relaxed, as linear identifiability seems to hold even when some of the assumptions are violated (cf. Tab. 5). In Appx. D, we demonstrate the remarkable robustness of latent identifiability (Fig. 6), the interaction between batch size, latent dimensionality, concentration, and latent conditional.

**Implications for Deep Learning.**  Our results indicate that deep learning models trained using cross-entropy and assuming a certain DGP recover the underlying latent variables up to linear transformations. As our identifiability proof for parametric instance discrimination illustrates with DIET, this statement also holds when the classification task is standard supervised learning. Our analysis on the key role of cross-entropy-based classification provides a theoretical foundation for phenomena such as neural analogy-making, transfer learning, and linear decoding of features.

**Conclusion.**  We extend the identifiability results of the auxiliary-variable nonlinear Independent Component Analysis (ICA) literature to parametric instance discrimination with a cluster-centric data generating process. Our modeling choice can capture the clustered structure of the data, accommodates non-normalized (as in ICA) and unit-normalized (as in InfoNCE) representations (Thm. 1). Furthermore, our identifiability result holds for the latent representation used post-training, i.e., for the latent variables before the classification head. Our results offer new insights into the success of deep learning, particularly in supervised classification tasks, which we show is a special case of the DIET parametric instance discrimination algorithm, where the instance labels equal the semantic class labels (Thm. 2). By linking self-supervised learning—via nonlinear ICA and DIET—to supervised classification for a specific DGP, we provide a theoretical framework that explains why simple classification tasks recover interpretable and transferable representations.

**Future Work.**  Future research could extend these insights to connections between nonlinear ICA and other forms of supervised learning and testing the scalability of our theoretical results to larger models and datasets. To assess our theory's predictions beyond proxy labels (Idrissi et al., 2022), we need real world image datasets with full specification of the latent variables, e.g., in rendered scenes.

ACKNOWLEDGMENTS

The authors thank the International Max Planck Research School for Intelligent Systems (IMPRS-IS) for supporting Patrik Reizinger and Attila Juhos. Patrik Reizinger acknowledges his membership in the European Laboratory for Learning and Intelligent Systems (ELLIS) PhD program. This work was supported by the German Federal Ministry of Education and Research (BMBF): Tübingen AI Center, FKZ: 01IS18039A. Wieland Brendel acknowledges financial support via an Emmy Noether Grant funded by the German Research Foundation (DFG) under grant no. BR 6382/1-1 and via the Open Philantropy Foundation funded by the Good Ventures Foundation. Wieland Brendel is a member of the Machine Learning Cluster of Excellence, EXC number 2064/1 – Project number 390727645. This research utilized compute resources at the Tübingen Machine Learning Cloud, DFG FKZ INST 37/1057-1 FUGG. Alice Bizeul's work is supported by an ETH AI Center Doctoral fellowship.

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

# A  IDENTIFIABILITY OF LATENTS DRAWN FROM A vMF AROUND CLUSTER VECTORS

This section contains the formal statement and proof of our main theoretical result. Appx. A.1 contains the relevant definition of affine generator systems. Appx. A.2 contains the assumptions and the proof for all four combinations of unit-normalized and non-normalized features/cluster vectors for parametric instance discrimination. Appx. A.3 discusses a special case, supervised classification.

## A.1  AFFINE GENERATOR SYSTEMS

**Definition 1** (Affine Generator System). *A system of vectors $\{\boldsymbol{v}_c \in \mathbb{R}^d | c \in \mathscr{C}\}$ is called an* affine generator system *if any vector in $\mathbb{R}^d$ is an affine linear combination of the vectors in the system. Put into symbols: for any $\boldsymbol{v} \in \mathbb{R}^d$ there exist coefficients $\alpha_c \in \mathbb{R}$, such that*

$$\boldsymbol{v} = \sum_{c \in \mathscr{C}} \alpha_c \boldsymbol{v}_c \quad and \quad \sum_{c \in \mathscr{C}} \alpha_c = 1. \tag{3}$$

**Lemma 1** (Properties of affine generator systems). *The following hold for any affine generator system $\{\boldsymbol{v}_c \in \mathbb{R}^d | c \in \mathscr{C}\}$:*

*1. for any $a \in \mathscr{C}$ the system $\{\boldsymbol{v}_c - \boldsymbol{v}_a | c \in \mathscr{C}\}$ is now a generator system of $\mathbb{R}^d$;*
*2. the invertible linear image of an affine generator system is also an affine generator system.*

## A.2  IDENTIFIABILITY OF PARAMETRIC INSTANCE DISCRIMINATION

**Assumptions 1C** (DGP with vMF samples around cluster vectors). *Assume the following DGP:*

  (i) *There exists a finite set of classes $\mathscr{C}$, represented by a set of unit-norm $d$-dimensional cluster-vectors $\{\boldsymbol{v}_c | c \in \mathscr{C}\} \subseteq \mathbb{S}^{d-1}$ such that they form an affine generator system of $\mathbb{R}^d$.*
 (ii) *There is a finite set of instance labels $\mathscr{I}$ and a well-defined, surjective class function $\mathcal{C} : \mathscr{I} \to \mathscr{C}$ (every label belongs to exactly one class and every class is in use).*
(iii) *A data sample $\boldsymbol{x}$ belongs to class $C = \mathcal{C}(I)$ and is labeled with a uniformly-chosen instance label, i.e., $I \in Uni(\mathscr{I})$.*
 (iv) *The latent $\boldsymbol{z} \in \mathbb{S}^{d-1}$ of our data sample with label $I$ is drawn from a vMF distribution around the cluster vector $\boldsymbol{v}_C$, where $C = \mathcal{C}(I)$:*

$$\boldsymbol{z} \sim p(\boldsymbol{z}|C) \propto e^{\kappa \langle \boldsymbol{v}_C, \boldsymbol{z} \rangle}. \tag{4}$$

  (v) *The data sample $\boldsymbol{x}$ is generated by passing the latent $\boldsymbol{z}$ through a continuous and injective generator function $\boldsymbol{g} : \mathbb{S}^{d-1} \to \mathbb{R}^D$, i.e., $\boldsymbol{x} = \boldsymbol{g}(\boldsymbol{z})$.*

Assume that, using the DIET objective (6), we train a continuous encoder $\boldsymbol{f} : \mathbb{R}^D \to \mathbb{R}^d$ on $\boldsymbol{x}$ and a linear classification head $\boldsymbol{W}$ on top of $\boldsymbol{f}$. The rows of $\boldsymbol{W}$ are $\{\boldsymbol{w}_i^\top | i \in \mathscr{I}\}$. In other words, $\boldsymbol{W}$ computes similarities (scalar products) between its rows and the embeddings:

$$\boldsymbol{W} : \boldsymbol{f}(\boldsymbol{x}) \mapsto \left[ \langle \boldsymbol{w}_i, \boldsymbol{f}(\boldsymbol{x}) \rangle | _{i \in \mathscr{I}} \right]. \tag{5}$$

In DIET, we optimize the following objective among all possible continuous encoders $\boldsymbol{f}$, linear classifiers $\boldsymbol{W}$, and $\beta > 0$:

$$\mathcal{L}(\boldsymbol{f}, \boldsymbol{W}, \beta) = \mathbb{E}_{(\boldsymbol{x}, I)} \left[ -\ln \frac{e^{\beta \langle \boldsymbol{w}_I, \boldsymbol{f}(\boldsymbol{x}) \rangle}}{\sum_{j \in \mathscr{I}} e^{\beta \langle \boldsymbol{w}_j, \boldsymbol{f}(\boldsymbol{x}) \rangle}} \right] \tag{6}$$

In the special case where the embeddings $\boldsymbol{f}(\boldsymbol{x})$ are unnormalized, but the parameter vectors $\boldsymbol{w}_i$ are unit-normalized, the identifiability proof will solicit another, technical assumption:

**Assumption 2** (Diverse data). *The system $\{\boldsymbol{v}_c | c \in \mathscr{C}\}$ is said to be diverse enough, if the following $|\mathscr{C}| \times 2d$ matrix has full column rank of $2d$:*

$$\begin{pmatrix} \cdots\cdots\cdots & \cdots\cdots\cdots \\ (\boldsymbol{v}_c \odot \boldsymbol{v}_c)^\top & \boldsymbol{v}_c^\top \\ \cdots\cdots\cdots & \cdots\cdots\cdots \end{pmatrix}, \tag{7}$$

*where $[\boldsymbol{x} \odot \boldsymbol{y}]_i = x_i y_i$ is the elementwise- or Hadamard product.*

*As long as $|\mathscr{C}| \geq 2d$, this property holds almost surely w.r.t. the Lebesgue-measure of $\mathbb{S}^{d-1}$ or any continuous probability distribution of $\boldsymbol{v}_c \in \mathbb{S}^{d-1}$.*

**Theorem 1C** (Identifiability of latents drawn from a vMF around cluster vectors). *Let $(\boldsymbol{f}, \boldsymbol{W}, \beta)$ globally minimize the DIET objective (6) under Assums. 1C and the following additional constraints:*

*C1. both the embeddings $\boldsymbol{f}(\boldsymbol{x})$ and $\boldsymbol{w}_i$'s are unit-normalized. Then:*
  - *(a) $\boldsymbol{h} = \boldsymbol{f} \circ \boldsymbol{g}$ is orthogonal linear, i.e., the latents are identified up to an orthogonal linear transformation;*
  - *(b) $\boldsymbol{w}_i = \boldsymbol{h}(\boldsymbol{v}_{\mathcal{C}(i)})$ for any $i \in \mathcal{I}$, i.e., $\boldsymbol{w}_i$'s identify the cluster-vectors $\boldsymbol{v}_c$ up to the same orthogonal linear transformation;*
  - *(c) $\beta = \kappa$, the temperature of the vMF distribution is also identified.*

*C2. the embeddings $\boldsymbol{f}(\boldsymbol{x})$ are unit-normalized, the $\boldsymbol{w}_i$'s are unnormalized. Then:*
  - *(a) $\boldsymbol{h} = \boldsymbol{f} \circ \boldsymbol{g}$ is orthogonal linear;*
  - *(b) $\boldsymbol{w}_i = \frac{\kappa}{\beta} \boldsymbol{h}(\boldsymbol{v}_{\mathcal{C}(i)}) + \boldsymbol{\psi}$ for any $i \in \mathcal{I}$, where $\boldsymbol{\psi}$ is a constant vector independent of $i$.*

*C3. the embeddings $\boldsymbol{f}(\boldsymbol{x})$ are unnormalized, while the $\boldsymbol{w}_i$'s are unit-normalized. If the system $\{\boldsymbol{v}_c | c\}$ is diverse enough in the sense of Assum. 2, then:*
  - *(a) $\boldsymbol{w}_i = \mathcal{O}\boldsymbol{v}_{\mathcal{C}(i)}$, for any $i \in \mathcal{I}$, where $\mathcal{O}$ is orthogonal linear;*
  - *(b) $\boldsymbol{h} = \boldsymbol{f} \circ \boldsymbol{g} = \frac{\kappa}{\beta}\mathcal{O}$ with the same orthogonal linear transformation, but scaled with $\frac{\kappa}{\beta}$.*

*C4. neither the embeddings $\boldsymbol{f}(\boldsymbol{x})$ nor the rows of $\boldsymbol{W}$ are unit-normalized. Then:*
  - *(a) $\boldsymbol{h} = \boldsymbol{f} \circ \boldsymbol{g}$ is linear;*
  - *(b) $\boldsymbol{w}_i$ identifies $\boldsymbol{v}_{\mathcal{C}(i)}$ up to an affine linear transformation.*

*Furthermore, in all cases, the row vectors that belong to samples of the same class are equal, i.e., for any $i, j \in \mathcal{I}$, $\mathcal{C}(i) = \mathcal{C}(j)$ implies $\boldsymbol{w}_i = \boldsymbol{w}_j$.*

**Remark.** *In cases C2 and C4, the cluster vectors are unnormalized and, therefore, can absorb the temperature parameter $\beta$. Thus $\beta$ can be set to $1$ without loss of generality. In case C3, it is $\boldsymbol{f}$ that can absorb $\beta$.*

*Proof.* **Step 1: Deriving an equation characterizing the global optimizers of the objective.**

**Rewriting the objective in terms of latents:** we plug the expression $\boldsymbol{x} = \boldsymbol{g}(\boldsymbol{z})$ into the optimization objective (6) to express the dependence in terms of the latents $\boldsymbol{z}$:

$$\mathcal{L}(\boldsymbol{f}, \boldsymbol{W}, \beta) = \mathbb{E}_{(\boldsymbol{z}, I)}\left[-\ln \frac{e^{\beta\langle \boldsymbol{w}_I, \boldsymbol{f} \circ \boldsymbol{g}(\boldsymbol{z})\rangle}}{\sum_{j \in \mathcal{I}} e^{\beta\langle \boldsymbol{w}_j, \boldsymbol{f} \circ \boldsymbol{g}(\boldsymbol{z})\rangle}}\right] = \mathcal{L}_{\boldsymbol{z}}(\boldsymbol{f} \circ \boldsymbol{g}, \boldsymbol{W}, \beta), \tag{8}$$

where the optimization is still over $\boldsymbol{f}$ (and not $\boldsymbol{h} = \boldsymbol{f} \circ \boldsymbol{g}$).

We note that the generator $\boldsymbol{g}$ is, by assumption, continuously invertible on the *compact* set $\mathbb{S}^{d-1}$. Therefore, its image $\boldsymbol{g}(\mathbb{S}^{d-1})$ is compact, too, and its inverse $\boldsymbol{g}^{-1}$ is also continuous. By Tietze's extension theorem (Wikipedia, 2024b), $\boldsymbol{g}^{-1}$ can be continuously extended to a function $\boldsymbol{F} : \mathbb{R}^D \to \mathbb{S}^{d-1}$. Therefore, any continuous function $\boldsymbol{h} : \mathbb{S}^{d-1} \to \mathbb{R}^d$ can take the role of $\boldsymbol{f} \circ \boldsymbol{g}$ by substituting $\boldsymbol{f} = \boldsymbol{h} \circ \boldsymbol{F}$ continuous, since now $\boldsymbol{f} \circ \boldsymbol{g} = \boldsymbol{h} \circ (\boldsymbol{F} \circ \boldsymbol{g}) = \boldsymbol{h} \circ id_{\mathbb{S}^{d-1}} = \boldsymbol{h}$.

Hence, minimizing $\mathcal{L}_{\boldsymbol{z}}(\boldsymbol{f} \circ \boldsymbol{g}, \boldsymbol{W}, \beta)$ (and by extension $\mathcal{L}(\boldsymbol{f}, \boldsymbol{W}, \beta)$) for continuous $\boldsymbol{f}$ equates to minimizing $\mathcal{L}_{\boldsymbol{z}}(\boldsymbol{h}, \boldsymbol{W}, \beta)$ for continuous $\boldsymbol{h}$:

$$\mathcal{L}_{\boldsymbol{z}}(\boldsymbol{h}, \boldsymbol{W}, \beta) = \mathbb{E}_{(\boldsymbol{z}, I)}\left[-\ln \frac{e^{\beta\langle \boldsymbol{w}_I, \boldsymbol{h}(\boldsymbol{z})\rangle}}{\sum_{j \in \mathcal{I}} e^{\beta\langle \boldsymbol{w}_j, \boldsymbol{h}(\boldsymbol{z})\rangle}}\right]. \tag{9}$$

**Expressing the condition for global optimality of the objective:** We rewrite the objective (9) by 1) using the indicator variable $\delta_{I=i}$ of the event $\{I = i\}$ and 2) applying the law of total expectation:

$$\mathcal{L}_{\boldsymbol{z}}(\boldsymbol{h}, \boldsymbol{W}, \beta) = \mathbb{E}_{(\boldsymbol{z}, I)}\left[-\sum_{i \in \mathcal{I}} \delta_{I=i} \ln \frac{e^{\beta\langle \boldsymbol{w}_i, \boldsymbol{h}(\boldsymbol{z})\rangle}}{\sum_{j \in \mathcal{I}} e^{\beta\langle \boldsymbol{w}_j, \boldsymbol{h}(\boldsymbol{z})\rangle}}\right] \tag{10}$$

$$= \mathbb{E}_{\boldsymbol{z}}\left[\mathbb{E}_I\left[-\sum_{i \in \mathcal{I}} \delta_{I=i} \ln \frac{e^{\beta\langle \boldsymbol{w}_i, \boldsymbol{h}(\boldsymbol{z})\rangle}}{\sum_{j \in \mathcal{I}} e^{\beta\langle \boldsymbol{w}_j, \boldsymbol{h}(\boldsymbol{z})\rangle}} \,\bigg|\, \boldsymbol{z}\right]\right]. \tag{11}$$

Using the properties that $\mathbb{E}\big[A\,f(B)\big|B\big] = \mathbb{E}\big[A\big|B\big]f(B)$ and that $\mathbb{E}[\delta_{I=i}] = \mathbb{P}(I=i)$, we conclude that:

$$\mathcal{L}_{\boldsymbol{z}}(\boldsymbol{h}, \boldsymbol{W}, \beta) = \mathbb{E}_{\boldsymbol{z}}\left[-\sum_{i \in \mathscr{I}} \mathbb{E}_I\left[\delta_{I=i} \ln \frac{e^{\beta\langle \boldsymbol{w}_i, \boldsymbol{h}(\boldsymbol{z})\rangle}}{\sum_{j \in \mathscr{I}} e^{\beta\langle \boldsymbol{w}_j, \boldsymbol{h}(\boldsymbol{z})\rangle}}\,\bigg|\,\boldsymbol{z}\right]\right] \tag{12}$$

$$= \mathbb{E}_{\boldsymbol{z}}\left[-\sum_{i \in \mathscr{I}} \mathbb{E}_I\left[\delta_{I=i}\big|\boldsymbol{z}\right] \ln \frac{e^{\beta\langle \boldsymbol{w}_i, \boldsymbol{h}(\boldsymbol{z})\rangle}}{\sum_{j \in \mathscr{I}} e^{\beta\langle \boldsymbol{w}_j, \boldsymbol{h}(\boldsymbol{z})\rangle}}\right] \tag{13}$$

$$= \mathbb{E}_{\boldsymbol{z}}\left[-\sum_{i \in \mathscr{I}} \mathbb{P}(I=i|\boldsymbol{z}) \ln \frac{e^{\beta\langle \boldsymbol{w}_i, \boldsymbol{h}(\boldsymbol{z})\rangle}}{\sum_{j \in \mathscr{I}} e^{\beta\langle \boldsymbol{w}_j, \boldsymbol{h}(\boldsymbol{z})\rangle}}\right]. \tag{14}$$

By Gibbs' inequality (Wikipedia, 2024a), the cross-entropy inside the expectation is globally minimized if and only if

$$\frac{e^{\beta\langle \boldsymbol{w}_i, \boldsymbol{h}(\boldsymbol{z})\rangle}}{\sum_{j \in \mathscr{I}} e^{\beta\langle \boldsymbol{w}_j, \boldsymbol{h}(\boldsymbol{z})\rangle}} = \mathbb{P}(I=i|\boldsymbol{z}), \quad \text{for any } i \in \mathscr{I}. \tag{15}$$

Moreover, the entire expectation is globally minimized if and only if the above equality (15) holds almost everywhere for $\boldsymbol{z} \in \mathbb{S}^{d-1}$.

Using that instance label $I$ is uniformly distributed, or $\mathbb{P}(I=j) = \mathbb{P}(I=i)$, the likelihood of the sample being in class $i$ can be expressed via Bayes' theorem as:

$$\mathbb{P}(I=i|\boldsymbol{z}) = \frac{p(\boldsymbol{z}|I=i)\mathbb{P}(I=i)}{\sum_{j \in \mathscr{I}} p(\boldsymbol{z}|I=j)\mathbb{P}(I=j)} = \frac{p(\boldsymbol{z}|I=i)}{\sum_{j \in \mathscr{I}} p(\boldsymbol{z}|I=j)}. \tag{16}$$

Substituting (16) into (15) yields that for any $i \in \mathscr{I}$ and almost everywhere w.r.t. $\boldsymbol{z} \in \mathbb{S}^{d-1}$:

$$\frac{e^{\beta\langle \boldsymbol{w}_i, \boldsymbol{h}(\boldsymbol{z})\rangle}}{\sum_{j \in \mathscr{I}} e^{\beta\langle \boldsymbol{w}_j, \boldsymbol{h}(\boldsymbol{z})\rangle}} = \frac{p(\boldsymbol{z}|I=i)}{\sum_{j \in \mathscr{I}} p(\boldsymbol{z}|I=j)}. \tag{17}$$

We now divide the equation (17) for the probability of a sample having label $i$ with that of having label $k$ and take the logarithm. This yields that $\mathcal{L}_{\boldsymbol{z}}(\boldsymbol{h}, \boldsymbol{W}, \beta)$ is globally minimized if and only if

$$\beta\langle \boldsymbol{w}_i - \boldsymbol{w}_k, \boldsymbol{h}(\boldsymbol{z})\rangle = \ln \frac{p(\boldsymbol{z}|I=i)}{p(\boldsymbol{z}|I=k)} \tag{18}$$

holds for any $i, k \in \mathscr{I}$ and almost everywhere w.r.t. $\boldsymbol{z} \in \mathbb{S}^{d-1}$.

**Plugging in the vMF distribution:** Plugging the assumed conditional distribution from (4) into (18) yields the equivalent expression:

$$\beta\langle \boldsymbol{w}_i - \boldsymbol{w}_k, \boldsymbol{h}(\boldsymbol{z})\rangle = \kappa\langle \boldsymbol{v}_{\mathcal{C}(i)} - \boldsymbol{v}_{\mathcal{C}(k)}, \boldsymbol{z}\rangle, \tag{19}$$

which holds for any $i, k \in \mathscr{I}$ and almost everywhere w.r.t. $\boldsymbol{z} \in \mathbb{S}^{d-1}$. Since $\boldsymbol{h}$ is continuous, the equation holds almost everywhere w.r.t. $\boldsymbol{z}$ if and only if it holds for all $\boldsymbol{z} \in \mathbb{S}^{d-1}$.

Observe that if $\boldsymbol{h} = id|_{\mathbb{S}^{d-1}}, \boldsymbol{w}_i = \boldsymbol{v}_{\mathcal{C}(i)}$ for any $i \in \mathscr{I}$, and $\beta = \kappa$, then the equation is satisfied. Thus, we can conclude that the global minimum of the cross-entropy loss is achieved.

**Step 2: Solving the equation for $h, W$ and proving identifiability.**

We now find all solutions to prove the identifiability of the latent variables and that of the cluster vectors. Denote $\tilde{\boldsymbol{w}}_i = \frac{\beta}{\kappa}\boldsymbol{w}_i$ to simplify the above equation to:

$$\langle \tilde{\boldsymbol{w}}_i - \tilde{\boldsymbol{w}}_k, \boldsymbol{h}(\boldsymbol{z})\rangle = \langle \boldsymbol{v}_{\mathcal{C}(i)} - \boldsymbol{v}_{\mathcal{C}(k)}, \boldsymbol{z}\rangle. \tag{20}$$

**$h$ is injective and has full-dimensional image:** We prove that $\boldsymbol{h}$ is injective. Assume that $\boldsymbol{h}(\boldsymbol{z}_1) = \boldsymbol{h}(\boldsymbol{z}_2)$ for some $\boldsymbol{z}_1, \boldsymbol{z}_2 \in \mathbb{S}^{d-1}$. Plugging $\boldsymbol{z}_1$ and $\boldsymbol{z}_2$ into (20) and subtracting the two equations yields:

$$0 = \langle \tilde{\boldsymbol{w}}_i - \tilde{\boldsymbol{w}}_k, \boldsymbol{h}(\boldsymbol{z}_1) - \boldsymbol{h}(\boldsymbol{z}_2)\rangle = \langle \boldsymbol{v}_{\mathcal{C}(i)} - \boldsymbol{v}_{\mathcal{C}(k)}, \boldsymbol{z}_1 - \boldsymbol{z}_2\rangle, \tag{21}$$

for any $i, k$. However, as the cluster vectors $\{\boldsymbol{v}_c|c\}$ form an affine generator system, the vectors $\{\boldsymbol{v}_{\mathcal{C}(i)} - \boldsymbol{v}_{\mathcal{C}(k)}|i, k\}$ form a generator system of $\mathbb{R}^d$ (see Lem. 1). Therefore, $\langle \boldsymbol{y}, \boldsymbol{z}_1 - \boldsymbol{z}_2\rangle = 0$, for any $\boldsymbol{y} \in \mathbb{R}^d$, which holds if and only if $\boldsymbol{z}_1 = \boldsymbol{z}_2$. Hence, $\boldsymbol{h}$ is injective.

By the Borsuk-Ulam theorem, for any continuous map from $\mathbb{S}^{d-1}$ to a space of dimensionality at most $d-1$ there exists some pair of antipodal points that are mapped to the same point. Consequently, no such function can be injective at the same time. Since $\boldsymbol{h} : \mathbb{S}^{d-1} \to \mathbb{R}^d$ is injective, the linear span of its image must be $\mathbb{R}^d$.

**Collapse of $w_i$'s:**  We prove that $\tilde{w}_i = \tilde{w}_k$ if $\mathcal{C}(i) = \mathcal{C}(k)$, i.e., samples from the same cluster will have equal rows of $W$ associated with them.

Assume that $\mathcal{C}(i) = \mathcal{C}(k)$ and substitute them into (20):

$$\langle \tilde{w}_i - \tilde{w}_k, h(z) \rangle = 0 \quad \text{for any } z \in \mathbb{S}^{d-1}. \tag{22}$$

However, we have just seen that the linear span of the image of $h$ is $\mathbb{R}^d$, which implies that $\tilde{w}_i = \tilde{w}_k$. We may abuse our notation by setting $\tilde{w}_c = \tilde{w}_i$ if $\mathcal{C}(i) = c$, which yields a new form for (20):

$$\langle \tilde{w}_a - \tilde{w}_b, h(z) \rangle = \langle v_a - v_b, z \rangle, \tag{23}$$

for any $a, b \in \mathscr{C}$ and any $z \in \mathbb{S}^{d-1}$.

**Linear transformation from $v_a - v_b$ to $\tilde{w}_a - \tilde{w}_b$:**  We now prove the existence of a linear map $\mathcal{A}$ on $\mathbb{R}^d$ such that $\mathcal{A}(v_a - v_b) = \tilde{w}_a - \tilde{w}_b$ for any $a, b \in \mathscr{C}$. For this, we prove that the following mapping is well-defined:

$$\mathcal{A} : \sum_{a,b \in \mathscr{C}} \lambda_{ab}(v_a - v_b) \mapsto \sum_{a,b \in \mathscr{C}} \lambda_{ab}(\tilde{w}_a - \tilde{w}_b). \tag{24}$$

Since the system $\{v_a - v_b | a, b\}$ is not necessarily linearly independent, we have to prove that the mapping is independent of the choice of the linear combination. More precisely if for some coefficients $\lambda_{ab}, \lambda'_{ab}$

$$\sum_{a,b \in \mathscr{C}} \lambda_{ab}(v_a - v_b) = \sum_{a,b \in \mathscr{C}} \lambda'_{ab}(v_a - v_b) \tag{25}$$

holds, then it should be implied that

$$\sum_{a,b \in \mathscr{C}} \lambda_{ab}(\tilde{w}_a - \tilde{w}_b) = \sum_{a,b \in \mathscr{C}} \lambda'_{ab}(\tilde{w}_a - \tilde{w}_b). \tag{26}$$

Assume that (25) holds. Then, the difference of the two sides is:

$$0 = \sum_{a,b \in \mathscr{C}} (\lambda_{ab} - \lambda'_{ab})(v_a - v_b). \tag{27}$$

Taking the scalar product with an arbitrary $z \in \mathbb{S}^{d-1}$ and using the linearity of the scalar product gives us:

$$0 = \langle \sum_{a,b \in \mathscr{C}} (\lambda_{ab} - \lambda'_{ab})(v_a - v_b), z \rangle = \sum_{a,b \in \mathscr{C}} (\lambda_{ab} - \lambda'_{ab})\langle v_a - v_b, z \rangle. \tag{28}$$

Now using (23) yields:

$$0 = \sum_{a,b \in \mathscr{C}} (\lambda_{ab} - \lambda'_{ab})\langle \tilde{w}_a - \tilde{w}_b, h(z) \rangle = \langle \sum_{a,b \in \mathscr{C}} (\lambda_{ab} - \lambda'_{ab})(\tilde{w}_a - \tilde{w}_b), h(z) \rangle. \tag{29}$$

However, the linear span of the image of $h$ is $\mathbb{R}^d$, which implies that

$$\sum_{a,b \in \mathscr{C}} (\lambda_{ab} - \lambda'_{ab})(\tilde{w}_a - \tilde{w}_b) = 0, \tag{30}$$

equivalent to (26). Therefore, the mapping is well-defined and the linearity of $\mathcal{A}$ follows.

**$h$ is linear:**  Equation (23) becomes:

$$\langle \mathcal{A}(v_a - v_b), h(z) \rangle = \langle v_a - v_b, z \rangle, \tag{31}$$

for any $a, b \in \mathscr{C}$ and any $z \in \mathbb{S}^{d-1}$. Nevertheless, $\{v_a - v_b | a, b \in \mathscr{C}\}$ is a generator system of $\mathbb{R}^d$, and, hence, (31) is equivalent to

$$\langle \mathcal{A}y, h(z) \rangle = \langle y, z \rangle, \quad \text{for any } y \in \mathbb{R}^d \text{ and any } z \in \mathbb{S}^{d-1}. \tag{32}$$

This is further equivalent to

$$\langle y, \mathcal{A}^\top h(z) \rangle = \langle y, z \rangle. \tag{33}$$

Since $y$ is arbitrary, we conclude that $\mathcal{A}^\top h(z) = z$ for any $z \in \mathbb{S}^{d-1}$. Therefore $\mathcal{A}$ is an invertible transformation and $h = (\mathcal{A}^\top)^{-1}$ is linear.

**Proving Thm. 1C case C4:** We have shown that $\boldsymbol{h}$ is linear. Furthermore, from (31) it follows, by fixing $b$ and defining $\boldsymbol{\psi} = \mathcal{A}\boldsymbol{v}_b - \boldsymbol{w}_b$, that

$$\tilde{\boldsymbol{w}}_a = \mathcal{A}\boldsymbol{v}_a + \boldsymbol{\psi}, \quad \text{for any } a \in \mathscr{C}, \tag{34}$$

which proves case C4 of Thm. 1C.

**Proving Thm. 1C case C2:** As a special case of the previous one, now we assume that $\boldsymbol{h}(\boldsymbol{z})$ is unit-normalized and maps $\mathbb{S}^{d-1}$ to $\mathbb{S}^{d-1}$. That amounts to $\boldsymbol{h} = (\mathcal{A}^\top)^{-1}$ being linear, norm-preserving, and therefore orthogonal. Consequently $\mathcal{A}$ is also orthogonal, $\boldsymbol{h} = \mathcal{A}$ and (34) simplifies to $\frac{\beta}{\kappa}\boldsymbol{w}_a = \tilde{\boldsymbol{w}}_a = \mathcal{A}\boldsymbol{v}_a + \boldsymbol{\psi} = \boldsymbol{h}(\boldsymbol{v}_a) + \boldsymbol{\psi}$, which proves C2 of Thm. 1C.

**Proving Thm. 1C case C1:** We now assume that both $\boldsymbol{h}$ and $\boldsymbol{w}_i$'s are unit-normalized. Consequently, $\boldsymbol{h} = \mathcal{A}$ is orthogonal linear and $\boldsymbol{w}_a = \frac{\kappa}{\beta}\mathcal{A}\boldsymbol{v}_a + \boldsymbol{\psi}$.

Therefore, on one hand, the $\boldsymbol{w}_a$'s lie on a $d$-dimensional hypersphere of radius $\frac{\kappa}{\beta}$ and center $\boldsymbol{\psi}$. On the other hand, by definition, $\boldsymbol{w}_a$'s also lie on the unit hypersphere $\mathbb{S}^{d-1}$.

Since the system $\{\boldsymbol{w}_a | a \in \mathscr{C}\}$ is the bijective affine linear image of the affine generator system $\{\boldsymbol{v}_a | a \in \mathscr{C}\}$, $\{\boldsymbol{w}_a | a \in \mathscr{C}\}$ is also an affine generator system (Lem. 1). Consequently, there could be at most one hypersphere in $\mathbb{R}^d$ which contains all the $\boldsymbol{w}_a$'s. Hence $\frac{\kappa}{\beta} = 1$, $\boldsymbol{\psi} = \boldsymbol{0}$, and $\boldsymbol{w}_a = \boldsymbol{h}(\boldsymbol{v}_a)$, which proves C1 of Thm. 1C.

**Proving Thm. 1C case C3:** Finally, we assume that $\boldsymbol{w}_i$'s are unit-normalized. As this is a special case of Thm. 1C C4, we know that there exists a constant vector $\boldsymbol{\psi}$ such that:

$$\boldsymbol{w}_a = \frac{\kappa}{\beta}\mathcal{A}\boldsymbol{v}_a + \boldsymbol{\psi}, \tag{35}$$

for any $a \in \mathscr{C}$. We are going to prove that $\mathcal{O} = \frac{\kappa}{\beta}\mathcal{A}$ is orthogonal and $\boldsymbol{\psi} = \boldsymbol{0}$.

Let $\mathcal{O} = \mathcal{U}^\top \Sigma \mathcal{V}$ be the singular value decomposition (SVD) of $\mathcal{O}$. Premultiplying with $\mathcal{U}$ yields:

$$\mathcal{U}\boldsymbol{w}_a = \Sigma \mathcal{V}\boldsymbol{v}_a + \mathcal{U}\boldsymbol{\psi}. \tag{36}$$

As orthogonal transformations $\mathcal{U}$ and $\mathcal{V}$ keep their arguments unit-normalized and $\{\mathcal{V}\boldsymbol{v}_a - \mathcal{V}\boldsymbol{v}_b\}$ is still an affine generator system (Lem. 1), we may assume without the loss of generality that

$$\boldsymbol{w}_a = \Sigma \boldsymbol{v}_a + \boldsymbol{\psi}, \tag{37}$$

for any $a \in \mathscr{C}$, where all $\boldsymbol{v}_a$'s and $\boldsymbol{w}_a$'s are unit-normalized.

Let us assume that $\boldsymbol{\psi} \neq \boldsymbol{0}$. In that case both sides of (37) can be scaled such that the offset $\boldsymbol{\psi}$ has unit norm. In this case $\boldsymbol{w}_a$'s are no longer on the unit hypersphere, but they instead have a mutual norm $r$. Assuming that the diagonal elements of $\Sigma$ are $\boldsymbol{\sigma} = (\sigma_1, \dots, \sigma_d)$, this is equivalent to:

$$r^2 = \|\Sigma \boldsymbol{v}_a + \boldsymbol{\psi}\|^2 = \|\Sigma \boldsymbol{v}_a\|^2 + 2\langle \Sigma \boldsymbol{v}_a, \boldsymbol{\psi}\rangle + \|\boldsymbol{\psi}\|^2 \tag{38}$$
$$= \langle \boldsymbol{v}_a \odot \boldsymbol{v}_a, \boldsymbol{\sigma} \odot \boldsymbol{\sigma}\rangle + \langle \boldsymbol{v}_a, 2\boldsymbol{\sigma} \odot \boldsymbol{\psi}\rangle + 1, \tag{39}$$

where $[\boldsymbol{x} \odot \boldsymbol{y}]_i = x_i y_i$ is the elementwise product. Eq. (39) is equivalent to the following:

$$(\boldsymbol{v}_a \odot \boldsymbol{v}_a)^\top (\boldsymbol{\sigma} \odot \boldsymbol{\sigma}) + \boldsymbol{v}_a^\top (2\boldsymbol{\sigma} \odot \boldsymbol{\psi}) - r^2 = -1. \tag{40}$$

Collecting the equations for all $a \in \mathscr{C}$ yields:

$$\mathcal{D}\begin{pmatrix} \boldsymbol{\sigma} \odot \boldsymbol{\sigma} \\ 2\boldsymbol{\sigma} \odot \boldsymbol{\psi} \\ r^2 \end{pmatrix} = -\mathbf{1}_{|\mathscr{C}|}, \tag{41}$$

where $\mathcal{D}$ is the following $|\mathscr{C}| \times (2d+1)$ matrix:

$$\mathcal{D} = \begin{pmatrix} \dots\dots\dots & \dots\dots\dots & \dots \\ (\boldsymbol{v}_a \odot \boldsymbol{v}_a)^\top & \boldsymbol{v}_a^\top & -1 \\ \dots\dots\dots & \dots\dots\dots & \dots \end{pmatrix}. \tag{42}$$

By Assum. 2, the left $|\mathscr{C}| \times 2d$ submatrix of $\mathcal{D}$ has full rank of $2d$. Consequently, the solution space to the more general, linear equation $\mathcal{D}\boldsymbol{t} = -\mathbf{1}_{|\mathscr{C}|}$, $\boldsymbol{t} \in \mathbb{R}^d$, has a dimensionality of at most 1. By the unit-normality of $\boldsymbol{v}_a$, we have $(\boldsymbol{v}_a \odot \boldsymbol{v}_a)^\top \mathbf{1}_d = 1$. From this, the solutions are exactly the following:

$$\boldsymbol{t} = \begin{pmatrix} \gamma \cdot \mathbf{1}_d \\ \mathbf{0}_d \\ \gamma + 1 \end{pmatrix}, \quad \text{where } \gamma \in \mathbb{R}. \tag{43}$$

Therefore, for any solution of (41) there exists $\gamma$ such that:

$$\boldsymbol{\sigma} \odot \boldsymbol{\sigma} = \gamma \cdot \mathbf{1}_d \tag{44}$$

$$\boldsymbol{\sigma} \odot \boldsymbol{\psi} = \mathbf{0}_d. \tag{45}$$

However, as the original transformation $\mathcal{A}$ was invertible, all singular values $\sigma_i$ are strictly positive and, thus, it follows that $\boldsymbol{\psi} = \mathbf{0}$. This is a technical contradiction to our initial assumption that $\boldsymbol{\psi} \neq \mathbf{0}$. Thus, it follows that $\boldsymbol{\psi} = \mathbf{0}$.

Therefore, (37) becomes:

$$\boldsymbol{w}_a = \Sigma \boldsymbol{v}_a, \tag{46}$$

where all $\boldsymbol{v}_a$'s and $\boldsymbol{w}_a$'s are unit-normalized. Following the same derivation yields:

$$1 = \|\Sigma \boldsymbol{v}_a\|^2 = (\boldsymbol{v}_a \odot \boldsymbol{v}_a)^\top (\boldsymbol{\sigma} \odot \boldsymbol{\sigma}), \tag{47}$$

or, after collecting the equations for all $a \in \mathscr{C}$:

$$\mathcal{B}(\boldsymbol{\sigma} \odot \boldsymbol{\sigma}) = \mathbf{1}_{|\mathscr{C}|}, \tag{48}$$

where $\mathcal{B}$ is the $|\mathscr{C}| \times d$ matrix

$$\mathcal{B} = \begin{pmatrix} \cdots\cdots\cdots \\ (\boldsymbol{v}_a \odot \boldsymbol{v}_a)^\top \\ \cdots\cdots\cdots \end{pmatrix}. \tag{49}$$

By Assum. 2, $\mathcal{B}$ has full rank, thus, there is at most one solution to the equation $\mathcal{B}\boldsymbol{t} = \mathbf{1}_{|\mathscr{C}|}$. Due to the unit-normality of $\boldsymbol{v}_a$'s, this solution is exactly $\boldsymbol{t} = \mathbf{1}_d$. However, as the singular values $\sigma_i$ are all positive, the only solution to $\boldsymbol{\sigma} \odot \boldsymbol{\sigma} = \mathbf{1}_d$ is $\boldsymbol{\sigma} = \mathbf{1}_d$. Equivalently, $\mathcal{O} = \frac{\kappa}{\beta}\mathcal{A}$ is orthogonal.

Furthermore, $\boldsymbol{h} = (\mathcal{A}^\top)^{-1} = (\frac{\beta}{\kappa}\mathcal{O}^\top)^{-1} = \frac{\kappa}{\beta}\mathcal{O}$. $\qquad\square$

### A.3 IDENTIFIABILITY OF SUPERVISED CLASSIFICATION

**Assumption 3** (DGP with vMF samples around cluster vectors). *Assume the following DGP:*

(i) *There exists a finite set of classes $\mathscr{C}$, represented by a set of unit-norm $d$-dimensional cluster-vectors $\{\boldsymbol{v}_c | c \in \mathscr{C}\} \subseteq \mathbb{S}^{d-1}$ such that they form an affine generator system of $\mathbb{R}^d$.*

(ii) *A data sample $\boldsymbol{x}$ belongs to a uniformly chosen class $C \in Uni(\mathscr{C})$.*

(iii) *The latent $\boldsymbol{z} \in \mathbb{S}^{d-1}$ of our data sample $\boldsymbol{x}$ with label $C$ is drawn from a vMF distribution around the cluster vector $\boldsymbol{v}_C$:*

$$\boldsymbol{z} \sim p(\boldsymbol{z}|C) \propto e^{\kappa\langle\boldsymbol{v}_C, \boldsymbol{z}\rangle}. \tag{50}$$

(iv) *The data sample $\boldsymbol{x}$ is generated by passing the latent $\boldsymbol{z}$ through a continuous and injective generator function $\boldsymbol{g} : \mathbb{S}^{d-1} \to \mathbb{R}^D$, i.e., $\boldsymbol{x} = \boldsymbol{g}(\boldsymbol{z})$.*

We would like to point out that the assumption of the class label $C$ being uniform restricts the scope the following theorem as it cannot account for imbalanced class labels. This shortcoming did not affect Thm. 1, as the uniform distribution over instance labels is a natural choice in practical scenarios with finite datasets.

**Theorem 2C** (Identifiability of latent variables drawn from a vMF around class vectors). *Let Assums. 1C hold and suppose that a continuous encoder $\boldsymbol{f} : \mathbb{R}^D \to \mathbb{R}^d$, a linear classifier $\boldsymbol{W}$ with rows $\{\boldsymbol{w}_c^\top \mid c \in \mathscr{C}\}$, and $\beta > 0$ globally minimize the cross-entropy objective:*

$$\mathcal{L}(\boldsymbol{f}, \boldsymbol{W}, \beta) = \mathbb{E}_{(\boldsymbol{x},C)}\left[-\ln \frac{e^{\beta\langle\boldsymbol{w}_C, \boldsymbol{f}(\boldsymbol{x})\rangle}}{\sum_{c' \in \mathscr{C}} e^{\beta\langle\boldsymbol{w}_{c'}, \boldsymbol{f}(\boldsymbol{x})\rangle}}\right].$$

*Then, the composition $\boldsymbol{h} = \boldsymbol{f} \circ \boldsymbol{g}$ is a linear map from $\mathbb{S}^{d-1}$ to $\mathbb{R}^d$.*

*Proof.*

**Step 1: Rewriting the Objective in Terms of $h$.** We begin by expressing the loss function in terms of the latent variable $z$. Recall that $x = g(z)$ and $h = f \circ g$. Substituting into the loss function:

$$\mathcal{L}(f, W, \beta) = \mathbb{E}_{(z,C)} \left[ -\ln \frac{e^{\beta \langle w_C, h(z) \rangle}}{\sum_{c' \in \mathscr{C}} e^{\beta \langle w_{c'}, h(z) \rangle}} \right]. \tag{51}$$

Since $g$ is continuous and injective on the compact set $\mathbb{S}^{d-1}$, its inverse $g^{-1}$ exists and is continuous on $g(\mathbb{S}^{d-1})$. By Tietze's extension theorem, we can extend $g^{-1}$ to a continuous function $g_{\text{ext}}^{-1} : \mathbb{R}^D \to \mathbb{S}^{d-1}$. Therefore, any continuous function $h : \mathbb{S}^{d-1} \to \mathbb{R}^d$ corresponds to a continuous encoder $f = h \circ g_{\text{ext}}^{-1}$, satisfying $f(x) = h(z)$.

**Step 2: Optimality Condition of the Cross-Entropy Loss.** At the global minimum of the cross-entropy loss, the predicted class probabilities match the true conditional probabilities almost everywhere. That is, for all $z \in \mathbb{S}^{d-1}$ and all $c \in \mathscr{C}$:

$$\frac{e^{\beta \langle w_c, h(z) \rangle}}{\sum_{c' \in \mathscr{C}} e^{\beta \langle w_{c'}, h(z) \rangle}} = \mathbb{P}(C = c \mid z). \tag{52}$$

**Step 3: Expressing the True Conditional Probabilities.** Using Bayes' theorem and the fact that classes are uniformly distributed ($\mathbb{P}(C = c)$ is constant[2]), we have:

$$\mathbb{P}(C = c \mid z) = \frac{p(z \mid C = c)}{\sum_{c' \in \mathscr{C}} p(z \mid C = c')}. \tag{53}$$

Given that, by assumption, the latent $z$ follows a von Mises-Fisher (vMF) distribution centered at $v_c$:

$$p(z \mid C = c) \propto e^{\kappa \langle v_c, z \rangle}. \tag{54}$$

Substituting into the conditional probability:

$$\mathbb{P}(C = c \mid z) = \frac{e^{\kappa \langle v_c, z \rangle}}{\sum_{c' \in \mathscr{C}} e^{\kappa \langle v_{c'}, z \rangle}}. \tag{55}$$

**Step 4: Equating Predicted and True Probabilities.** Setting the predicted probabilities equal to the true probabilities, we obtain:

$$\frac{e^{\beta \langle w_c, h(z) \rangle}}{\sum_{c' \in \mathscr{C}} e^{\beta \langle w_{c'}, h(z) \rangle}} = \frac{e^{\kappa \langle v_c, z \rangle}}{\sum_{c' \in \mathscr{C}} e^{\kappa \langle v_{c'}, z \rangle}}. \tag{56}$$

Dividing the expressions for classes $c$ and $c'$, we eliminate the denominators:

$$\frac{e^{\beta \langle w_c, h(z) \rangle}}{e^{\beta \langle w_{c'}, h(z) \rangle}} = \frac{e^{\kappa \langle v_c, z \rangle}}{e^{\kappa \langle v_{c'}, z \rangle}}. \tag{57}$$

Taking the logarithm of both sides:

$$\beta \left( \langle w_c, h(z) \rangle - \langle w_{c'}, h(z) \rangle \right) = \kappa \left( \langle v_c, z \rangle - \langle v_{c'}, z \rangle \right). \tag{58}$$

Simplifying:

$$\beta \langle w_c - w_{c'}, h(z) \rangle = \kappa \langle v_c - v_{c'}, z \rangle. \tag{59}$$

---

[2]We acknowledge that this assumption does not hold in many realistic scenarios, where the data distribution is unbalanced between the classes

**Step 5: Defining Scaled Parameters.** Let us define:

$$\tilde{w}_c = \frac{\beta}{\kappa} \boldsymbol{w}_c. \tag{60}$$

Then the key equation becomes:

$$\langle \tilde{w}_c - \tilde{w}_{c'}, \boldsymbol{h}(\boldsymbol{z}) \rangle = \langle \boldsymbol{v}_c - \boldsymbol{v}_{c'}, \boldsymbol{z} \rangle, \quad \forall c, c' \in \mathscr{C}. \tag{61}$$

**Step 6: Establishing a Linear Relationship.** Define the difference vectors:

$$\delta_{\tilde{w}_{cc'}} = \tilde{w}_c - \tilde{w}_{c'}, \quad \delta_{\boldsymbol{v}_{cc'}} = \boldsymbol{v}_c - \boldsymbol{v}_{c'}. \tag{62}$$

Our key equation is now:

$$\langle \delta_{\tilde{w}_{cc'}}, \boldsymbol{h}(\boldsymbol{z}) \rangle = \langle \delta_{\boldsymbol{v}_{cc'}}, \boldsymbol{z} \rangle, \quad \forall c, c' \in \mathscr{C}. \tag{63}$$

Since the set $\{\delta_{\boldsymbol{v}_{cc'}} \mid c, c' \in \mathscr{C}\}$ spans $\mathbb{R}^d$ (due to the affine generator system property), we can interpret this equation as stating that the inner products between $\boldsymbol{h}(\boldsymbol{z})$ and $\delta_{\tilde{w}_{cc'}}$ correspond to the inner products between $\boldsymbol{z}$ and $\delta_{\boldsymbol{v}_{cc'}}$.

**Step 7: Proving Injectivity and Full Rank of $\boldsymbol{h}$.** Suppose there exist $\boldsymbol{z}_1, \boldsymbol{z}_2 \in \mathbb{S}^{d-1}$ such that $\boldsymbol{h}(\boldsymbol{z}_1) = \boldsymbol{h}(\boldsymbol{z}_2)$. Then, for all $c, c' \in \mathscr{C}$:

$$\langle \delta_{\boldsymbol{v}_{cc'}}, \boldsymbol{z}_1 - \boldsymbol{z}_2 \rangle = \langle \delta_{\tilde{w}_{cc'}}, \boldsymbol{h}(\boldsymbol{z}_1) - \boldsymbol{h}(\boldsymbol{z}_2) \rangle = 0. \tag{64}$$

Since $\{\delta_{\boldsymbol{v}_{cc'}}\}$ spans $\mathbb{R}^d$, it follows that $\boldsymbol{z}_1 - \boldsymbol{z}_2 = \boldsymbol{0}$, i.e., $\boldsymbol{z}_1 = \boldsymbol{z}_2$. Therefore, $\boldsymbol{h}$ is injective.

By the Borsuk-Ulam theorem, an injective continuous map from $\mathbb{S}^{d-1}$ to $\mathbb{R}^{d'}$ with $d' < d$ cannot exist. Thus, the image of $\boldsymbol{h}$ must be full-dimensional in $\mathbb{R}^d$.

**Step 8: Defining a Linear Map $\mathcal{A}$.** We aim to find a linear map $\mathcal{A} : \mathbb{R}^d \to \mathbb{R}^d$ such that:

$$\delta_{\tilde{w}_{cc'}} = \mathcal{A}^\top \delta_{\boldsymbol{v}_{cc'}}, \quad \forall c, c' \in \mathscr{C}. \tag{65}$$

This is well-defined because any linear dependency among the $\delta_{\boldsymbol{v}_{cc'}}$ translates to the same linear dependency among the $\delta_{\tilde{w}_{cc'}}$, as shown below.
Suppose there are scalars $\{\lambda_{cc'}\}$ such that:

$$\sum_{c,c'} \lambda_{cc'} \delta_{\boldsymbol{v}_{cc'}} = \boldsymbol{0}. \tag{66}$$

Then, using the key equation (63):

$$\sum_{c,c'} \lambda_{cc'} \langle \delta_{\tilde{w}_{cc'}}, \boldsymbol{h}(\boldsymbol{z}) \rangle = \sum_{c,c'} \lambda_{cc'} \langle \delta_{\boldsymbol{v}_{cc'}}, \boldsymbol{z} \rangle = \left\langle \sum_{c,c'} \lambda_{cc'} \delta_{\boldsymbol{v}_{cc'}}, \boldsymbol{z} \right\rangle = 0. \tag{67}$$

Since $\boldsymbol{h}$ is injective and its image spans $\mathbb{R}^d$, the only way for this to hold for all $\boldsymbol{h}(\boldsymbol{z})$ is if:

$$\sum_{c,c'} \lambda_{cc'} \delta_{\tilde{w}_{cc'}} = \boldsymbol{0}. \tag{68}$$

Therefore, $\mathcal{A}^\top$ is a well-defined linear map.

**Step 9: Concluding that $h$ is Linear.**   Using the linear map $\mathcal{A}^\top$, the key equation becomes:

$$\langle \mathcal{A}^\top \delta_{\boldsymbol{v}_{cc'}}, \boldsymbol{h}(\boldsymbol{z}) \rangle = \langle \delta_{\boldsymbol{v}_{cc'}}, \boldsymbol{z} \rangle. \tag{69}$$

This implies:

$$\langle \delta_{\boldsymbol{v}_{cc'}}, \mathcal{A}\boldsymbol{h}(\boldsymbol{z}) - \boldsymbol{z} \rangle = 0, \quad \forall c, c' \in \mathscr{C}. \tag{70}$$

Since $\{\delta_{\boldsymbol{v}_{cc'}}\}$ spans $\mathbb{R}^d$, it follows that:

$$\mathcal{A}\boldsymbol{h}(\boldsymbol{z}) = \boldsymbol{z}, \quad \forall \boldsymbol{z} \in \mathbb{S}^{d-1}. \tag{71}$$

Therefore, $\boldsymbol{h}$ is the inverse of $\mathcal{A}$ restricted to $\mathbb{S}^{d-1}$, and since $\mathcal{A}$ is linear and invertible (due to the injectivity of $\boldsymbol{h}$), it follows that $\boldsymbol{h}$ is linear:

$$\boldsymbol{h}(\boldsymbol{z}) = \mathcal{A}^{-1}\boldsymbol{z}. \tag{72}$$

This completes the proof that $\boldsymbol{h}$ is linear.

**Step 10: Conclusion.**   Under the given assumptions, we have shown that $\boldsymbol{h} = \boldsymbol{f} \circ \boldsymbol{g}$ must be a linear function. This means that the latent variables $\boldsymbol{z}$ are identifiable up to a linear transformation determined by $\mathcal{A}^{-1}$.

$\square$

## B   THE GENEALOGY OF CROSS-ENTROPY–BASED CLASSIFICATION METHODS

This section provides the necessary background on auxiliary-variable ICA and discusses the connection between ICA and DIET, and InfoNCE and DIET.

### B.1   AUXILIARY-VARIABE NONLINEAR ICA: GENERALIZED CONTRASTIVE LEARNING (GCL)

In this section, we discuss the most general auxiliary-variable nonlinear ICA, termed Generalized Contrastive Learning (GCL) (Hyvarinen et al., 2019). GCL uses a conditionally factorizing source distribution (given auxiliary variable $u$): $\log p(\mathbf{s}|u)$ is a sum of components $q_i(s_i, u)$:

$$\log p(\mathbf{s}|u) = \sum_i q_i(s_i, u) \tag{73}$$

For this generalized model, Hyvarinen et al. (2019) define the following variability condition:

**Assumption 4** (Assumption of Variability). *For any $\mathbf{y} \in \mathbb{R}^n$ (used as a drop-in replacement for the sources $\mathbf{s}$), there exist $2n + 1$ values for the auxiliary variable $\mathbf{u}$, denoted by $\mathbf{u}_j, j = 0 \ldots 2n$ such that the $2n$ vectors in $\mathbb{R}^{2n}$ given by*

$$(\mathbf{w}(\mathbf{y}, \mathbf{u}_1) - \mathbf{w}(\mathbf{y}, \mathbf{u}_0)), (\mathbf{w}(\mathbf{y}, \mathbf{u}_2) - \mathbf{w}(\mathbf{y}, \mathbf{u}_0)) \ldots, (\mathbf{w}(\mathbf{y}, \mathbf{u}_{2n}) - \mathbf{w}(\mathbf{y}, \mathbf{u}_0))$$

*with*

$$\mathbf{w}(\mathbf{y}, \mathbf{u}) = \left( \frac{\partial q_1(y_1, \mathbf{u})}{\partial y_1}, \ldots, \frac{\partial q_n(y_n, \mathbf{u})}{\partial y_n}, \frac{\partial^2 q_1(y_1, \mathbf{u})}{\partial y_1^2}, \ldots, \frac{\partial^2 q_n(y_n, \mathbf{u})}{\partial y_n^2} \right)$$

*are linearly independent.*

Assum. 4 constrains the components of the first- and second derivatives of the functions constituting the sources' conditional log-density, given the auxiliary variable $\mathbf{u}$. As the authors write: *"[Assum. 4] is basically saying that the auxiliary variable must have a sufficiently strong and diverse effect on the distributions of the independent components."*

We state the required assumptions for the identifiability of GCL, adapted from (Hyvarinen et al., 2019, Thm. 1):

**Assumption 5** (Auxiliary-variable ICA with conditionally independent sources (GCL)). *We assume the following for latent factors $\boldsymbol{z}$, observations $\boldsymbol{x}$, generative model $\boldsymbol{g}$, encoder $\boldsymbol{f}$ (parametrized by a neural network), linear map $\boldsymbol{W}$ with $(\boldsymbol{f}, \boldsymbol{W})$ solving a multinomial regression problem:*

1. *The observations are generated with a diffeomorphism $\boldsymbol{g}: \boldsymbol{x} = \boldsymbol{g}(\boldsymbol{z})$, where $\dim \boldsymbol{x} = \dim \boldsymbol{z} = d$*
2. *The source components $z_i$ are conditionally independent, given a fully observed, $m-$dimensional random variable (RV) $\mathbf{u}$, i.e.,*

$$\log p(\boldsymbol{z}|\mathbf{u}) = \sum_i q_i(z_i, \mathbf{u}), \tag{74}$$

3. *The conditional log-pdf $q_i$ is sufficiently smooth as a function of $z_i$ for any fixed $\mathbf{u}$*
4. *Assum. 6 holds*
5. *the multinomial regression function*

$$r(\boldsymbol{x}, \mathbf{u}) = \sum_i^n \psi_i(f_i(\boldsymbol{x}), \mathbf{u}), \tag{75}$$

*discriminating $(\boldsymbol{x}, \mathbf{u})$ vs $(\boldsymbol{x}, \mathbf{u}^*)$ has universal approximation capability, both for $\psi_i$ and a diffeomorphic $\boldsymbol{f} = (f_1, \ldots, f_n)$ (parametrized by a neural network)*

When Assum. 5 holds, Hyvarinen et al. (2019) showed identifiability up to component-wise invertible transformations.

For the special case when the conditional distribution comes from the exponential family (in the case of our chosen vMF conditional, the distribution has order one), Assum. 5 turns into a simpler form (Assum. 6).

## B.2 PARAMETRIC INSTANCE DISCRIMINATION (DIET) AND TIME-CONTRASTIVE LEARNING (TCL)

**Arbitrary labels: time and sample index** As Hyvarinen et al. (2019) note in (Hyvarinen et al., 2019, 5.4), $\mathbf{u}$ can stand for many types of additional information. TCL uses the time index, which is assumed to be a RV. Importantly, an arbitrarily defined class label, such as in DIET, can serve the same purpose. In this case, we denote the auxiliary variable $u = c$

**Adapting the assumptions between TCL and DIET.** The only reason we cannot apply (Hyvarinen et al., 2019, Thm. 1) is that our exponential family has order one, violating Assum. 4. This fact, however, shows our theory's consistency as we cannot go beyond identifiability up to linear (orthogonal or affine) transformation.

To fit our theory into the ICA family of methods, we note that modeling the DGP in DIET with a cluster-centric approach, we naturally fit most of the ICA assumptions. To compare our Assums. 1C to all the assumptions used for (Hyvarinen et al., 2019, Thm. 3) (cf. Assum. 5), we note that the vMF distribution belongs to the exponential family, and that requiring that the cluster vectors form an affine generator system (cf. Appx. A.1 for a definition and properties) satisfies the special case of the general sufficient variability Assum. 4 condition:

**Assumption 6** (Sufficient variability). *Define the modulation parameter matrix $\mathbf{L} \in \mathbb{R}^{(E-1) \times dk}$ for $d-$dimensional exponential family distributions of order $k$ with rows as:*

$$[\mathbf{L}]_{j:} = (\boldsymbol{\theta}^j - \boldsymbol{\theta}^0)^\top \tag{76}$$

$$\boldsymbol{\theta}^j = [\theta_{11}^j, \ldots, \theta_{dk}^j]. \tag{77}$$

*Then, sufficient variability means that $\operatorname{rank}(\mathbf{L}) = dk$, i.e., the modulation parameter matrix has full **column** rank.*

To see how a vMF fulfills Assum. 6, consider that the log-pdf $q_i(z_i, c)$ comes from a conditional exponential family, i.e.:

$$q_i(z_i, c) = \sum_{j=1}^k [\tilde{q}_{ij}(z_i)\,\theta_{ij}(c)] - \log N_i(c) + \log Q_i(z_i), \tag{78}$$

$$= \kappa \langle \boldsymbol{v}_c, \boldsymbol{z} \rangle + \log C_d(\kappa) \tag{79}$$

where $k$ is the order of the exponential family, $N_i$ is the normalizing constant, $Q_i$ the base measure, $\tilde{q}_i$ is the sufficient statistics, and the modulation parameters $\theta_i := \theta_i(c)$ depend on $c$. In our cluster-centric vMF conditional in (2), $k = 1$ (i.e., we can drop the $j$ index) and $\theta_i(c) = \boldsymbol{v}_c$. This corresponds to (79) above, where $C_d(\kappa)$ is the concentration- and dimension-dependent normalization constant.

As our DGP assumes that the cluster vectors form an affine generator system, and in the above Eq. (78) the cluster vectors take the role of $\theta_{ij}(c)$, we can prove that our DGP fulfils Assum. 6.

**Lemma 2** (The cluster-based DIET DGP is sufficiently variable). *Assuming that the cluster-vectors form an affine generator system (Assums. 1C), then the modulation parameter matrix $\mathbf{L}$ (defined in Assum. 6) formed by the cluster vectors $\boldsymbol{v}_c - \boldsymbol{v}_a$ has full column rank.*

*Proof.* First we need to show that the cluster vectors $\boldsymbol{v}_c$ have the same role as $\theta_{ij}(c)$. The derivative of the log-pdf of the vMF distribution in (79) w.r.t. $\boldsymbol{z}$ is the exponent in the DIET conditional (we can differentiate for non-normalized $\boldsymbol{z}$, which is the case for auxiliary-bvariable ICA).

$$\frac{\partial}{\partial \boldsymbol{z}}[\kappa\langle\boldsymbol{v}_c, \boldsymbol{z}\rangle + \log C_d(\kappa)] = \kappa\boldsymbol{v}_c \tag{80}$$

Then, we need $d + 1$ cluster vectors to use one as a pivot to calculate $\mathbf{L}$ as defined in Assum. 6. By Lem. 1, this new set of vectors (i.e., offset by $\boldsymbol{v}_a$, expressed as $\boldsymbol{v}_c - \boldsymbol{v}_a$) also forms a generator system of $\mathbb{R}^d$, which implies that $\mathbf{L}$ has rank $d$, concluding the proof. $\square$

To apply (Hyvarinen et al., 2019, Thm. 3) to recover the identifiability result of TCL, we need to show that our setting can solve the regression problem defined in Assum. 5. What we will show, w.l.o.g., is that our regression function akin to (Hyvarinen et al., 2019, (11)) does not have an auxiliary-variable dependent constant.

**Lemma 3** (Regression function). *The regression function in (Hyvarinen et al., 2019, Thm. 3), which solves the multiclass classification problem, consists of three items: 1) a scalar product of vector-valued functions of either $\boldsymbol{z}$ or c, and scalar-valued functions of 2) $\boldsymbol{z}$ and 3) c. Our DGP and neural network pipeline used for learning can also match this regression function, by choosing a pivot (zero) value for the c-dependent scalar function. This is without loss of generality.*

*Proof.* In Thm. 1, the identifiability of the cluster vectors is up to an affine transformation, where the bias is denoted by $\boldsymbol{\psi}$. Calculating the scalar product of the learned cluster vector with the learned latents yields two terms:

1. a scalar product term between $\boldsymbol{z}-$ and $c-$dependent vectors; and
2. a $\boldsymbol{\psi} \cdot \boldsymbol{h}(\boldsymbol{z})$ term, which depends only on $\boldsymbol{z}$

Comparing to (Hyvarinen et al., 2019, Eq. (11)), we see that a $c-$dependent scalar function is missing. Following the common practice in multinomial regression, we can, w.l.o.g., arbitrarily choose the pivot value of the $c-$dependent scalar function to be 0, thus we do not need that term. This yields the following expression for the regression function:

$$r(\boldsymbol{x}, c) = \boldsymbol{h}(\boldsymbol{x})^\top \tilde{\boldsymbol{w}}_c = \boldsymbol{h}(\boldsymbol{x})^\top (\mathcal{A}\boldsymbol{v}_c + \boldsymbol{\psi}) = \boldsymbol{h}(\boldsymbol{x})^\top \mathcal{A}\boldsymbol{v}_c + \boldsymbol{h}(\boldsymbol{x})^\top \boldsymbol{\psi} \tag{81}$$

$$= \boldsymbol{h}(\boldsymbol{x})^\top \mathcal{A}\boldsymbol{v}_c + a(\boldsymbol{x}), \tag{82}$$

where $\boldsymbol{h}$ is linear, so the first term depends on both $\boldsymbol{x}$, c, the second term $a(\boldsymbol{x})$ only on $\boldsymbol{x}$, and we can choose (as usual practice in MLR), w.l.o.g., $b(c) = 0$, which concludes the proof. $\square$

## B.3 THE RELATIONSHIP BETWEEN INFONCE AND DIET

Last, we show how DIET relates to InfoNCE, where we reframe InfoNCE in form of instance disrimination. InfoNCE optimizes the cross entropy between the true conditional of the underlying DGP (a vMF distribution) and the approximate conditional parametrized by an encoder network. This cross entropy can be formulated as a loss for an $N-$class classification problem, where $N$ is the dataset size:

$$\mathcal{L} = \sum_{i=1}^{B} CE(q(\boldsymbol{x}_i^+), \mathbf{e}_i) \quad \text{s.t.} \quad q_k(\boldsymbol{x}_i^+) = \frac{\exp\left(\boldsymbol{f}(\boldsymbol{x}_i^+)^\top \boldsymbol{f}(\boldsymbol{x}_k)\right)}{\sum_{b=1...B} \exp\left(\boldsymbol{f}(\boldsymbol{x}_i^+)^\top \boldsymbol{f}(\boldsymbol{x}_b)\right)}, \tag{83}$$

where $\mathbf{e}_i$ is the $i^{th}$ unit vector, encoding the class label in a one-hot fashion, and $\boldsymbol{x}_i^+$ denotes the positive pairs. Note that the last part is simply the standard softmax $\sigma(.)$ over the innner product $(\boldsymbol{f}(\boldsymbol{x}_i^+)^\top \boldsymbol{f}(\boldsymbol{x}_k))$. To go from InfoNCE to DIET, we need to make the following modifications:

1. Sum over the whole dataset $N$, not just the batch $B$.

2. Replace the encoding of the anchor sample $\boldsymbol{f}(\boldsymbol{x}_k)$ with a learnable linear projection $\boldsymbol{W}$, i.e., setting $q(\boldsymbol{x}_i^+) = \sigma(\boldsymbol{W}\boldsymbol{f}(\boldsymbol{x}_i^+))$

A remaining difference to the original DIET formulation is that InfoNCE assumes unit-normalized features. However, our theory (cf. Thm. 1C) can accommodate unit-normalized vectors, so this is not a problem.

Let $(\boldsymbol{x}_n, \boldsymbol{x}_n^+)$ be positive pair for sample $n$ and let there be $N$ samples in total. The InfoNCE loss is equivalent to a multi-class $N-$pair loss of the form:

$$\mathcal{L} = \sum_{i=1}^{B} CE(q(\boldsymbol{x}_i^+), \mathbf{e}_i) \quad \text{s.t.} \quad q_k(\boldsymbol{x}_i^+) = \frac{\exp\left(\boldsymbol{f}(\boldsymbol{x}_i^+)^\top \boldsymbol{f}(\boldsymbol{x}_k)\right)}{\sum_{b=1\ldots B} \exp\left(\boldsymbol{f}(\boldsymbol{x}_i^+)^\top \boldsymbol{f}(\boldsymbol{x}_b)\right)}. \tag{84}$$

Now instead of having mini-batches of size $B$, we take the loss over the whole dataset:

$$\mathcal{L} = \sum_{i=1}^{N} CE(q(\boldsymbol{x}_i^+), \mathbf{e}_i) \quad \text{s.t.} \quad q_k(\boldsymbol{x}_i^+) = \frac{\exp\left(\boldsymbol{f}(\boldsymbol{x}_i^+)^\top \boldsymbol{f}(\boldsymbol{x}_k)\right)}{\sum_{b=1\ldots N} \exp\left(\boldsymbol{f}(\boldsymbol{x}_i^+)^\top \boldsymbol{f}(\boldsymbol{x}_b)\right)}. \tag{85}$$

Next, replace $\boldsymbol{f}(\boldsymbol{x}_k)$ with a learnt and normalized weight vector $\boldsymbol{w}_k$:

$$\mathcal{L} = \sum_{i=1}^{N} CE(q(\boldsymbol{x}_i^+), \mathbf{e}_i) \quad \text{s.t.} \quad q_k(\boldsymbol{x}_i^+) = \frac{\exp\left(\boldsymbol{f}(\boldsymbol{x}_i^+)^\top \boldsymbol{w}_k\right)}{\sum_{b=1\ldots N} \exp\left(\boldsymbol{f}(\boldsymbol{x}_i^+)^\top \boldsymbol{w}_b\right)}. \tag{86}$$

Note that the last part is simply the standard softmax $\sigma(.)$ over a linear projection:

$$\mathcal{L} = \sum_{i=1}^{N} CE(q(\boldsymbol{x}_i^+), \mathbf{e}_i) \quad \text{s.t.} \quad q(\boldsymbol{x}_i^+) = \sigma(\boldsymbol{W}\boldsymbol{f}(\boldsymbol{x}_i^+)) \tag{87}$$

where $\boldsymbol{W}$ is the projection matrix for which the $k^{th}$ row corresponds to $\boldsymbol{w}_k$. Since $i$ in this case corresponds to the sample index in the dataset, we recovered DIET up to normalization, and so $\boldsymbol{W}$ is simply the linear classifier.

## C    ADDITIONAL EXPERIMENTAL DETAILS

### C.1    SYNTHETIC DATA

The code is based on `https://brendel-group.github.io/cl-ica/`.

### C.2    DISLIB

We evaluate our methods on the DisLib disentanglement benchmark (Locatello et al., 2019), which provides a controlled setting for testing disentanglement and latent variable recovery. We used the version of the DisLib (Locatello et al., 2019) dataset based on the GitHub repository from (Roth et al., 2022)[3]. It includes the vision datasets dSprites, Shapes 3D, MPI 3D, Cars 3D, and smallNORB. Using Pytorch, we train both a three-layer MLP with $512$ latent dimensions and BatchNorm (which helped with trainability) and a CNN (ResNet18) also with $512$ latent dimensions (He et al., 2016). We only consider latent variables with Euclidean topology, as non-Euclidean, e.g., periodic latent variables such as orientation, are problematic to learn and are potentially mapped to a nonlinear manifold (Higgins et al., 2018; Pfau et al., 2020; Keurti et al., 2023; Engels et al., 2024). We evaluate the recovery of latent variables by computing the Pearson correlation between ground-truth and predicted factors. Both models were trained for 100 epochs, with the Adam optimizer, a learning rate of 0.001 and a batch size of 4096.

### C.3    REAL DATA: IMAGENET-X

Finally, we test the generalizability of our theoretical insights on real-world data using ImageNet-X (Idrissi et al., 2022). The latent variables are proxies, defined by human annotators (Idrissi et al., 2022). They are binary labels, representing the deviation of a certain latent variable on a given sample from the mode of that latent variable. We evaluate how well linear decoders can predict

---

[3]`https://github.com/facebookresearch/disentangling-correlated-factors`

latent variables from pretrained model representations. We use two architectures, a ResNet50 (latent dimension $d = 2048$) and a Vit-b-16 (latent dimension $d = 768$) both trained on standard supervised classification using a cross-entropy loss on the full ImageNet dataset (Deng et al., 2009). Moreover, to get a baseline decoding performance from inputs (like in the DisLib experiments), we also fix a random linear projection from the full $224 \cdot 224 \cdot 3 = 150,528$ ImageNet input dimensionality down to 2048 the ResNet50 latent dimensionality. This is purely for computational reasons and can be justified based on the Johnson–Lindenstrauss lemma[4].

We randomly split the data into 70% training and 30% testing data. For some latent variables, the label distribution was heavily imbalanced with less than 1% positive examples. To compensate class imbalance, for each latent variable, we resampled both training and testing data to achieve an even distribution. We repeat this and all following analysis averaged over 10 random seeds. Using the `LogisticRegression` module from `sklearn`[5], we fit a linear decoder to predict the latent variable. Finally, we compute $p$-values based on one sample $t$-tests against a null hypothesis of chance level (50%) accuracy with a multi-comparison Bonferroni adjusted significance level of $\alpha = \frac{0.05}{17 \cdot 5} < 0.0006$ (17 factors and 5 models).

## D  ADDITIONAL EXPERIMENTAL RESULTS

**Ablating the choice of the cluster vectors.**  In Tab. 5, we present additional ablation studies exploring the effect of varying the distribution of the cluster vectors $\boldsymbol{v}_c$ on the unit hyper-sphere. We do not observe any significant impact on the $R^2$ scores of more concentrated cluster centroids $\boldsymbol{v}_c$.

Table 5: Identifiability in the synthetic setup. Mean $\pm$ standard deviation across 5 random seeds. Settings that match our theoretical assumptions are ✓. We report the $R^2$ score for linear mappings, $\tilde{\boldsymbol{z}} \to \boldsymbol{z}$ and $\boldsymbol{w}_i \to \boldsymbol{v}_c$ for cases with normalized (o) and unormalized (a) $\boldsymbol{w}_i$. For unormalized $\boldsymbol{w}_i$, we verify that mappings $\tilde{\boldsymbol{z}} \to \boldsymbol{z}$ are orthogonal by reporting the mean absolute error between their singular values and those of an orthogonal transformation.

| | | | | | | normalized $\boldsymbol{w}_i$ cases | | | | unnormalized $\boldsymbol{w}_i$ | |
| | | | | | | $R^2_{\mathrm{o}}(\uparrow)$ | | $\mathrm{MAE}_{\mathrm{o}}(\downarrow)$ | | $R^2_{\mathrm{a}}(\uparrow)$ | |
| $N$ | $d$ | $|\mathscr{C}|$ | $p(\boldsymbol{v}_c)$ | $p(\boldsymbol{z}|\boldsymbol{v}_c)$ | M. | $\tilde{\boldsymbol{z}} \to \boldsymbol{z}$ | $\boldsymbol{w}_i \to \boldsymbol{v}_c$ | $\tilde{\boldsymbol{z}} \to \boldsymbol{z}$ | $\boldsymbol{w}_i \to \boldsymbol{v}_c$ | $\tilde{\boldsymbol{z}} \to \boldsymbol{z}$ | $\boldsymbol{w}_i \to \boldsymbol{v}_c$ |
|---|---|---|---|---|---|---|---|---|---|---|---|
| $10^3$ | 5 | 100 | Uniform | vMF($\kappa{=}10$) | ✓ | $98.6_{\pm 0.01}$ | $99.9_{\pm 0.01}$ | $0.01_{\pm 0.00}$ | $0.00_{\pm 0.00}$ | $99.0_{\pm 0.00}$ | $99.9_{\pm 0.00}$ |
| $10^3$ | 5 | 100 | Laplace | vMF($\kappa{=}10$) | ✓ | $98.7_{\pm 0.00}$ | $99.5_{\pm 0.00}$ | $0.01_{\pm 0.00}$ | $0.00_{\pm 0.00}$ | $99.1_{\pm 0.00}$ | $99.8_{\pm 0.00}$ |
| $10^3$ | 5 | 100 | Normal | vMF($\kappa{=}10$) | ✓ | $98.2_{\pm 0.01}$ | $99.2_{\pm 0.01}$ | $0.01_{\pm 0.00}$ | $0.00_{\pm 0.00}$ | $99.2_{\pm 0.00}$ | $99.8_{\pm 0.00}$ |

**Quantifying the violation of the assumption on the conditional with a generalized normal.** Tabs. 2 and 3 show that using a Laplace conditional instead of a vMF or normal distribution leads to substantially lower $R^2$ scores, though one might argue that the Laplace distribution is not that different (according to some intuitive notion) from the vMF or normal distributions. To understand why using a Laplace conditional leads to such a poor performance, we ran synthetic experiments with a generalized normal conditional with scale $\alpha$ (this is conceptually similar to our concentration parameter $\kappa$) and shape $\beta$:

$$\boldsymbol{z} \sim p(\boldsymbol{z}|C) \propto e^{\alpha \|\boldsymbol{v}_C - \boldsymbol{z}\|_\beta^\beta}, \text{ where } \|\boldsymbol{x}\|_\beta^\beta = \sum_{i=1}^d |x_i|^\beta. \tag{88}$$

Importantly, $\beta = 1$ gives a Laplace, whereas $\beta = 2$ gives a normal distribution. Thus, the generalized normal can be thought of as "interpolating" between these two distributions, providing the perfect testbed to determine when performance starts to break down. We show the $R^2$ scores for both recovering $\boldsymbol{z}$ (Fig. 4 Left) and $\boldsymbol{v}_c$ (Fig. 4 Middle) across multiple scale ($\alpha$) and shape ($\beta$) values, averaged over 5 seeds. We also report the average representation norm across multiple scale ($\alpha$) and

---

[4]`https://en.wikipedia.org/wiki/JohnsonLindenstrauss_lemma`
[5]`https://scikit-learn.org/1.5/modules/generated/sklearn.linear_model.`
`LogisticRegression.html`

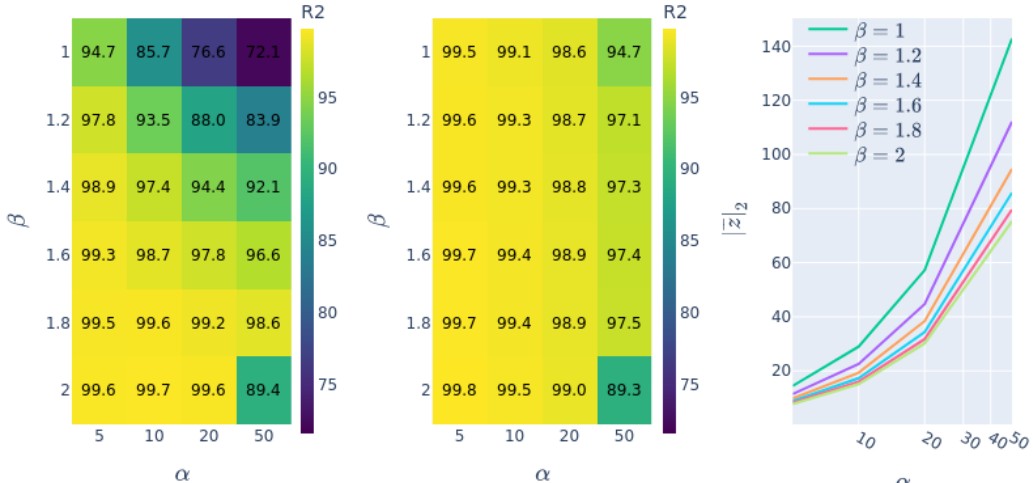

Figure 4: **Quantifying the assumption violation of a Laplace conditional:** Tabs. 2 and 3 show that using a Laplace conditional leads to substantially lower $R^2$ scores. Numbers are averages across 5 seeds. **(Left and Middle:)** using a generalized normal distribution to "interpolate" between a normal ($\beta = 2$) and a Laplace ($\beta = 1$) distribution for different scale values (denoted as $\alpha$, which is conceptually akin to our concentration $\kappa$) and show the $R^2$ score for recovering $\boldsymbol{z}$ **(Left)** and $\boldsymbol{v}_c$ **(Middle)**. **(Right:)** The average norm of the representation for the one-dimensional case for different $\beta$ values. As $\beta$ approaches 1, the average norm increases, indicating a larger spread

shape ($\beta$) values (calculated in the one-dimensional case and averaged over 5 seeds) and the crucial effect of having a fat tail (Fig. 5 Right) with a truncated Laplace distribution. Our results indicate that:

1. $\boldsymbol{v}_c$ **is easier to recover than $\boldsymbol{z}$:** the numbers are higher in Fig. 4(Middle) than in Fig. 4(Left)
2. **More concentrated conditionals degrade identifiability for all shapes:** $R^2$ scores decrease as $\alpha$ increases in both Fig. 4(Left, Middle)
3. **The average representation norm increases with increasing scale and decreasing shape:** as the conditional approaches the Laplace distribution $\beta \to 1$, the samples have a larger norm, i.e., they are further away from the unit hypersphere, potentially leading to insufficient overlap Fig. 4(Right)
4. **Fat tails worsen identifiability performance:** Allow more and more of the tail of a Laplace distribution to be included in the support (truncated symmetrically between $-1, 1$ and $-3, 3$ shows a strong anti-correlation with the $R^2$ score for multiple scales ($\alpha$).

**Loss saturation: the role of batch size and latent dimensionality.**    Our results in Tabs. 2 and 3 show that increasing latent dimensionality leads to substantially lower $R^2$ scores—in line with the findings in many prior works on the identifiability of SSL methods (Zimmermann et al., 2021; von Kügelgen et al., 2021; Rusak et al., 2024). In general, the issue of extracting large dimensional representations from practical, real-world datasets is an open question (Simon et al., 2023; Jing et al., 2022). We investigate the interaction between batch size and concentration in Fig. 5(Left).

With increasing concentrations, intra-cluster samples become more indistinguishable. This means that achieving close to optimum instance discrimination loss is easy with relatively coarse-grain features. The intuition is that most SSL objectives (including DIET) saturate, i.e.. get close to optimum as the underlying pretext problem (in our cases the instance discrimination) is nearly solved—a very good example is learning only the content features (but not the style ones) in von Kügelgen et al. (2021). The result is a population gradient with a very low norm. As the empirical loss is calculated based on a finite batch size, the variance of the gradient overtakes the norm, and the training effectively stalls. Additionally, higher concentrations result in less overlap between classes, which can have a detrimental effect on source recovery. However, the signal-to-noise ratio improves with a larger batch size, as the increasing $R^2$ scores show in Fig. 5(Left) in the columns from left to right. The results

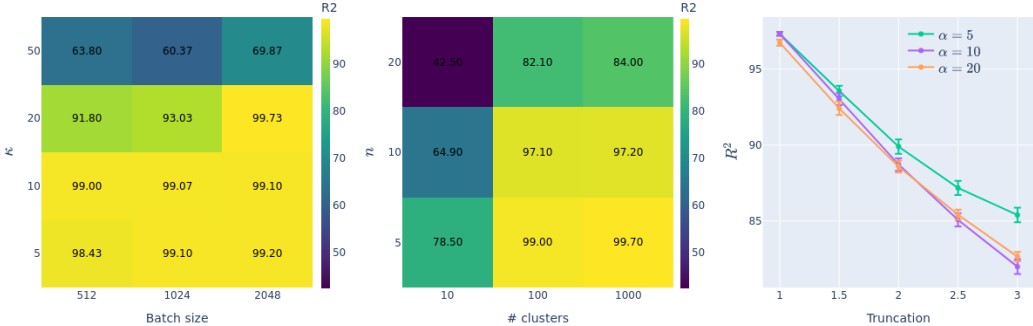

Figure 5: **The role of batch size, number of clusters, and fat tails for identifiability: (Left:)** Increasing batch size improves $R^2$ scores, counteracting the detrimental effect of more concentrated (higher $\kappa$) conditionals; **(Middle:)** More clusters improve $R^2$ scores, counteracting the detrimental effect higher dimensional representations (10 clusters for 10 and 20 dimensions violate Assum. 2; thus, the low $R^2$ score); **(Right:)** The Laplace distribution leads to low $R^2$ scores due to its fat tails. Experiments with a truncated Laplace conditional (where the support is restricted to $[-\text{Truncation}; \text{Truncation}]$) shows that the closer the truncated Laplace distribution is to the Laplace distribution (i.e., with increasing Truncation), $R^2$ scores decrease for all tested scales $\alpha$. Averages and error bars are reported across 5 seeds

suggest that this issue can, at least theoretically, be mitigated—note that identifiability results hold in the infinite-sample regime, so requiring a larger batch size does not contradict our results.

When the latent dimensionality increases, the saturating loss implies that relatively low-dimensional features are sufficient to achieve this near-optimum loss. This phenomenon can also be seen in Fig. 5(Middle), where for each column (i.e., a fixed number of clusters), the $R^2$ score deteriorates with increasing latent dimensionality.

It is important to mention that none of these contradict our theory, which holds for the global optimizer of the DIET population loss. In practice, the additional challenges of estimation error (due to finite sample size and finite batch size) and algorithmic error (using GD-based methods to solve a likely non-convex problem) may impose adverse effects on the evaluation.

**Diversity: the role of the number of clusters.** To investigate the role of Assum. 2, we investigate how the number of clusters affects the $R^2$ score. Though the number of clusters is intrinsic to the dataset, thus, it cannot be chosen arbitrarily, knowing its effect on performance can inform practitioners about potential failure cases. We ablated the number of clusters for different latent dimensionalities (Fig. 5(Middle)). For a given dimensionality, the $R^2$ score improves with more clusters, although there seems to be a sweet spot where a further increase in the number of clusters only marginally improves the $R^2$ score. We also see that our requirement of $d+1$ clusters (affine generator systems of $\mathbb{R}^d$ have at least this many members) is essential for good performance. This is reflected in the extremely poor $R^2$ scores for $n \in \{10, 20\}$, when the number of clusters is only 10 (first column from the left in Fig. 5(Middle)).

**Robustness to label noise.** In this section, we evaluate the robustness of our method under increasing label noise. For DIET (Fig. 6 Left) we perturbed the instance label for each sample with a probability equal to the label noise ratio ($x$-axis in the figure). The perturbed labels were drawn uniformly from the set of all instance labels. For the supervised case (Fig. 6 Right), the cluster label was perturbed instead. In both cases, the $y$-axis represents the identifiability score ($R^2$ score).

We believe this setup reflects realistic scenarios, where label noise is equally likely to affect any data point as commonly assumed in the literature (Nettleton et al., 2010; Frénay & Verleysen, 2013).

Despite this increasingly challenging setup, the results demonstrate remarkable robustness to label noise. Up to an 80% label noise ratio, latent recovery shows minimal degradation, and the cluster recovery performance of DIET remains perfect—though both metric substantially decrease for a larger (100%) ratio. We attribute this robustness to the symmetry of label noise across all instances and labels. While increased uncertainty does reduce the accuracy of individual label predictions, the

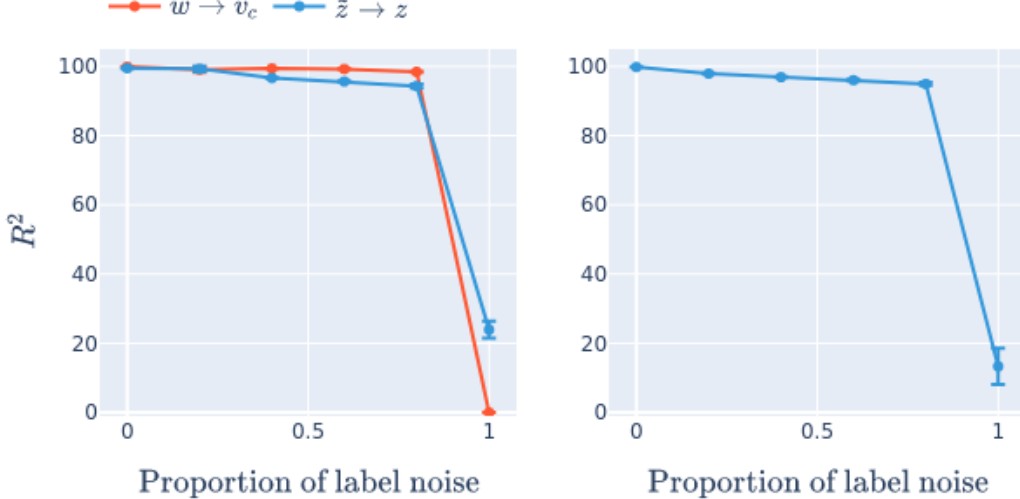

Figure 6: **The robustness of DIET and supervised classification to label noise:** The $x-$axis shows the proportion of instances with perturbed labels; the $y-$axis the $R^2$ score of learning the ground truth latents (and the cluster vectors on the left). Perturbed labels are uniformly resampled from the whole instance label (left) or cluster label (right) sets, respectively. **(Left:)** DIET perfectly recovers the cluster vectors $\boldsymbol{v}_c$ up to $80\%$ label noise, and shows only a small degradation for the latents $\boldsymbol{z}$; **(Right:)** Supervised classification robustly recovers the latents up to $80\%$ label noise with only a small degradation. Averages and error bars are reported across 5 seeds.

optimal logit values predicted by the encoder shouldn't change under symmetrical label noise. The stability of deep learning models to label noise has also been shown by Rolnick (2017).

## E ACRONYMS

**DGP** data generating process

**GCL** Generalized Contrastive Learning

**ICA** Independent Component Analysis

**LVM** latent variable model

**MAE** Mean Absolute Error

**PID** parametric instance discrimination

**RV** random variable

**SSL** self-supervised learning

**TCL** Time-Contrastive Learning

**vMF** von Mises-Fisher

## F NOMENCLATURE

$R^2$ coefficient of determination

$\mathcal{S}$ hypersphere

