# OpenReview forum: "Cross-Entropy Is All You Need To Invert the Data Generating Process"
_ICLR.cc/2025/Conference — ICLR 2025 Oral_

### Official Review · Reviewer_wFAY · 2024-10-25

**Soundness:** 3
**Presentation:** 2
**Contribution:** 3
**Rating:** 8
**Confidence:** 4

**Summary:**

In this work they show that under some assumptions on the distribution of the ground truth latent variables and the
distribution of the labels then a model trained with cross-entropy is able to recover the ground truth latents up to
a linear transformation.

**Strengths:**

S1: The result that under Assumptions 1C Appendix A.2 we get linear identifiable representations is both new and interesting.

**Weaknesses:**

W1: The article makes too strong a claim that cross-entropy on its own gives linearly identifiable representations.
This claim is too strong, because additional assumptions (Assumptions 1C Appendix A.2) are needed to get the result.
Places where this claim is made:
- The title
- The abstract: "We prove that even in standard classification tasks, models learn representations of ground-truth factors of variation up to a linear transformation."
- line 255-256: "Namely, supervised learning performs non-linear ICA."
- Line 304-305: "providing a theoretical explanation for the success of supervised learning in learning linearly decodable representations."
- Line 310-311: "Solving an (almost) arbitrary classification task by optimizing the cross entropy objective is sufficient to invert the DGP and identify the ground-truth representation up to a linear transformation"
- Line 521-525: The "Implications for Deep Learning" paragraph.
- Line 534-536: "By linking self-supervised learning—via nonlinear ICA and DIET—to supervised classification, we provide a theoretical framework that explains why simple classification tasks recover interpretable and transferable representations."


W2: The assumption made that "instance label I is uniformly distributed" (Assumptions 1C, (iii)) is important
because many classification tasks have unbalanced labels. Therefore, this assumption should be made more clear.


W3: In line 485, 518-519, you write: "Our experimental results also suggest that our assumptions can be relaxed,
as linear identifiability seems to hold even when some of the assumptions are violated". However, there are also
some examples in tables 2 and 3 of $R^2$ being surprisingly low even though all the assumptions should be satisfied.
The authors should be more clear about where this discrepancy comes from.

**Questions:**

Q1: In Table 2 with $N=10^3$, d=20, 100 classes, vMF($\kappa = 10$) and $N=10^3$, d=5, 100 classes, vMF($\kappa = 50$)
some of the $R^2$ values are much lower. You write that this is
"possibly explained by the content-style partitioning of latents (von Kügelgen et al., 2021) and insufficient
augmentation overlap (Wang et al., 2022; Rusak et al., 2024)." What do you mean by this?
I am guessing that you would give the same explanation for table 3, d=5, 100 classes, vMF($\kappa = 50$)?


Q2: In table 4, what is your interpretation of some of the pearson correlations being less than 0.8 and a few even
less than 0.6?

Q3: In fig. 3, most factors of variation are captured with better than chance performance, but a few are not and many
seem quite close to chance. Do you have an idea of why this might be?

---

> ### Author Response · Authors · 2024-11-22
>
> We thank Reviewer wFAY for praising our identifiability result for being **“both new and interesting”** and for the constructive feedback regarding phrasing and experimental evaluation.
>
> ### W1: Claim strength
> Thank you for pointing out the underspecification of our claim. We changed our framing in all text instances to include that besides a cross entropy-based objective, it is also crucial to assume a certain data generating process. We revised the manuscript accordingly.
>
> ### W2: Uniformly distributed instance labels
> Thank you for pointing this out. We agree that, for the supervised case, this assumption can often be unrealistic. We moved out the assumptions used for Thm. 2. into Assms. 3, where we clearly state and discuss this limitation.
>
> In the self-supervised instance discrimination case, the assumption that the instance label is uniformly chosen from the set of instances is a very natural one and doesn’t restrict the scope of Thm. 1. The instance label only refers to index of the sample within the training dataset and not to the actual class/cluster. In fact, the set of instances belonging to the individual clusters may be highly imbalanced and our theorem still applies.
>
> ### W3 and Q1: Explaining lower R2 scores for higher latent dimensionsionalities and conditional concentrations
> Thank you for pointing out the need to better explain why identifiability performance can suffer, even though our  assumptions are satisfied.
>
> The increasing difficulty of recovering higher dimensional ground truth latents is especially often encountered in prior works on identifiability under SSL methods [1,2,3]. In a DGP-agnostic setup, the issue of extracting large dimensional representations from practical, real-world datasets is also an open question (see e.g. [4, 5]).
>
> We speculate that a **major reason is the low signal-to-noise ratio of the gradient when computed on a finite batch size**. The intuition is that most SSL objectives (including DIET) saturate, i.e. get close to optimum as the underlying pretext problem (in our cases the instance discrimination) is nearly solved. However, relatively low-dimensional features are sufficient to achieve this near-optimum discrimination. Thus, the population gradient has a very low norm. Nevertheless, as the empirical loss is calculated based on a finite batch size, the variance of the gradient overtakes the norm, and the training effectively stalls.
>
> For example, with the higher concentration values intra-cluster samples become more indistinguishable. This implies that the uncertainty in predicting the correct instance label increases, i.e. the loss function has a very high optimum value. **Achieving close to optimum instance discrimination loss is easy with relatively coarse-grain features** that are sufficient for discerning samples coming from different clusters. In the revised manuscript, we **ablated the effect of high concentration under increasing batch size in Appx. D and Fig. 5 Left**. The results suggest that this issue can, at least theoretically, be mitigated. Appx. D also contains this discussion.
>
> For the **increased dimensionality $d=20$**, we also ran an ablation under increasing batch size. **With substantially increased training times, we saw surprising gains in recovery score across all batch sizes**. In the unnormalized $w_i$’s case, for dimensionality $d=20$ and batch size $1024$ we improved the identifiability score from the previous 81.9% (Table 2) up to 85.7%. Nevertheless, we could not yet achieve complete identifiability. Settings with much larger batch sizes (2048-65K) seem to benefit from the increased training time even later than the settings with smaller batch size. Due to this reason, we cannot yet confirm a scaling law of the $R^2$ score wrt batch size and we postpone the full ablation until post-rebuttal.
>
> It is important to mention that **none of these observations contradict our theory, as in the latter we made statements about the global optimizer of the DIET population loss**. In practice, the additional challenges of estimation error (due to finite sample size and finite batch size) and algorithmic error (using GD-based methods to solve a likely non-convex problem) may impose adverse effects on the evaluation.
>
> We note that Reviewer **vcau** had very similar questions. Thus, our updated submission provides an extensive empirical evaluation of different kinds of assumption violations in Appendix D, along the same line of reasoning.
>
> ### Bibliography
> [1] Zimmermann et al. (2021): “Contrastive Learning Inverts the Data Generating Process”
>
> [2] Von Kügelgen et al. (2021): “Self-Supervised Learning with Data Augmentations Provably Isolates Content from Style”
>
> [3] Rusak et al. (2024): “InfoNCE: Identifying the Gap Between Theory and Practice”
>
> [4] Simon et al. (2020): “On the Stepwise Nature of Self-Supervised Learning”
>
> [5] Jing et al. (2021): "Understanding Dimensional Collapse in Contrastive Self-supervised Learning"

---

> > ### Author Response · Authors · 2024-11-22
> >
> > ### Q2: Low Pearson correlations for DisLib
> > Good catch! The main reason we see mixed results on DisLib compared to our simulation experiments is that none of the datasets in Table 4 satisfy the assumptions of our data generating process. They are not constrained to a hypersphere and there is no clustering structure. In fact, the category labels are independent of all the other latents, rather than providing rich conditional distributions as we assume in our proof. The second reason is that all our results are asymptotic, we assume that our model can be trained to reach the global minimum of the loss function. However, in Table 4 we see that different architectures (MLP, CNN) achieve different performance levels (see correlation of category latent, which is the training target) due to their distinct inductive biases. Working towards a non-asymptotic theory of approximate identifiability presents an exciting future research avenue [6].
> >
> > ### Q3: Some factors are not recovered on ImageNet-X
> > Thank you for your question! We re-evaluated our results on ImageNet-X and now provide statistical tests to have a better overview on the performance of trained models in recovering (proxy) latent features. As our new Figure 3 suggests, almost all features are recovered better than chance performance (except “brighter” - second subplot in the second row) both for trained ViTs and ResNet50.
> >
> > ### Bibliography
> > [6] Lyu and Fu. (2022). “On finite-sample identifiability of contrastive learning-based nonlinear independent component analysis”. In International Conference on Machine Learning (pp. 14582-14600). PMLR.

---

> > > ### Comment · Reviewer_wFAY · 2024-11-25
> > >
> > > Thank you for the clarifications.
> > > I will keep my recommendation to accept the paper.

---

### Official Review · Reviewer_vcau · 2024-10-31

**Soundness:** 3
**Presentation:** 4
**Contribution:** 3
**Rating:** 8
**Confidence:** 4

**Summary:**

- This paper extends identifiability results for self-supervised learning techniques to include classifiers trained with cross entropy, under certain Assumptions.
- The Assumptions include a strong condition on the underlying data generating process that is unlikely to hold in practice: a clustering model with von Mises-Fisher samples around the cluster centers to yield the latent factors.
- Under their Assumptions, they present a theorem claiming that such cluster-center-plus-vMF latents are identifiable up to a linear transformation using the DIET objective (which is cross entropy where the labels are training sample indices).
- The paper validates the theoretical results on synthetic data where the latent factor is known, showing that in many of the settings covered by the theory, the R^2 score between the true and predicted latents are over 98%.
- It further explores empirical applicability of the theory on the DisLib, a disentanglement benchmark with known factors of variation. While the data generating process does not match with the theoretical model in this setting, they still demonstrate generally strong Pearson correlations between the learned representations and the factors of variation.
- A more challenging study using human-labeled factors of variation in the ImageNet-X dataset gives much weaker, but still positive, correlations between the factors and the learned representations.

**Strengths:**

- The paper is clear and well-written.
- The experiments are appropriate and demonstrate that the theoretical results at least occasionally apply beyond the setting of the fairly restrictive assumptions about the data generating process.

**Weaknesses:**

- Tables 2.3/Synthetic Data experiments: The identifiability suffers substantially as the dimensionality increases above 10, even if the perfect model is well-specified. This seems like potentially a major issue in practice, as the true latents of interest may often be very high-dimensional, and the model will often be misspecified. You acknowledge that this is an issue, but perhaps some additional discussion could be beneficial.
- I’m not sure that the two synthetic misspecification examples are very convincing about the broad generality of the theory. Using a Laplace distribution in place of the vMF degrades performance substantially already, and I’m not sure it should be considered a particularly bad case of misspecification. Perhaps including experiments with mixtures of Gaussians, with varying numbers of mixture components would help understand how well the theory holds up over more challenging misspecification settings?
- The results on ImageNet-X appear quite weak, which seems to indicate that violation of the Assumptions by the data generating process leads to substantial reduction in identifiability. Perhaps some additional discussion to help the reader understand why these results are weak for most of the factors of variation would help?
- Line 71: Extraneous comma after the hyphen.
- Line 457: “underlying training classification training task was solved” => “underlying training classification task was solved.” (Extra “training”, missing “.”)
- Line 527: “Conlusion” => “Conclusion”.

**Questions:**

- What would be the impact of repeated elements in the training data when using the sample index classification head? In this case, the representation would try to map one input to two (or more) arbitrary indices. Would this impact the identifiability results in any way?
- I’m not sure I understand the explanations for why a high concentration parameter is so detrimental to the identifiability. Could you give some additional intuition?
- Relatedly, high concentration doesn’t seem to violate any of the listed Assumptions (and you don’t mark it with a red X in the tables) – is this poor performance an indication of a missing Assumption?

---

> ### Author Response · Authors · 2024-11-22
>
> We thank Reviewer vcau for **praising our experimental evaluation**, including cases of assumption violations, and our submission’s **clear writing**. We are also grateful for pointing out unclear parts and for the constructive suggestions, which we answer in the following.
>
> ### W1 & Q2 & Q3: Declining identifiability with increasing dimensionality and concetration/Potential violation of our assumptions
> As the Reviewer noted, increasing the concentration parameter or the latent dimensionality is challenge in the empirical demonstration of the identifiability result.
>
> On one hand, the increasing difficulty of recovering higher dimensional ground truth latents is especially often encountered in prior works on identifiability under SSL methods [1,2,3]. On the other hand, in a DGP-agnostic setup, the issue of extracting large dimensional representations from practical, real-world datasets is also an open question (see e.g. [4, 5]).
>
> We speculate that a **major reason is the low signal-to-noise ratio of the gradient when computed on a finite batch size**. The intuition is that most SSL objectives (including DIET) saturate, i.e. get close to optimum as the underlying pretext problem (in our cases the instance discrimination) is nearly solved. However, relatively low-dimensional features are sufficient to achieve this near-optimum discrimination. Thus, the population gradient has a very low norm. Nevertheless, as the empirical loss is calculated based on a finite batch size, the variance of the gradient overtakes the norm, and the training effectively stalls.
>
> For example, with higher concentration values intra-cluster samples become more indistinguishable. This implies that the uncertainty in predicting the correct instance label increases, i.e. the loss function has a very high optimum value. **Achieving close to optimum instance discrimination loss is easy with relatively coarse-grain features** that are sufficient for discerning samples coming from different clusters. In the revised manuscript, we **ablated the effect of high concentration under increasing batch size in Appx. D and Fig. 5 Left**. The results suggest that this issue can, at least theoretically, be mitigated. Appx. D also contains this discussion.
>
> For the **increased dimensionality $d=20$**, we also ran an ablation under increasing batch size. **With substantially increased training times, we saw surprising gains in recovery score across all batch sizes**. For instance, in the unnormalized $w_i$’s case, for dimensionality $d=20$ and batch size $1024$ we managed to improve the identifiability score from the previous 81.9% (Table 2) up to 85.7%. Nevertheless, we could not yet achieve complete identifiability. Besides, settings with much larger batch sizes (2048-65K) seem to benefit from the increased training time even later than the settings with smaller batch size. Due to this reason, we cannot yet confirm a scaling law of the $R^2$ score wrt batch size and we postpone the full ablation until post-rebuttal.
>
> It is important to mention that **none of these observations contradict our theory, as in the latter we made statements about the global optimizer of the DIET population loss**. In practice, the additional challenges of estimation error (due to finite sample size and finite batch size) and algorithmic error (using GD-based methods to solve a likely non-convex problem) may impose adverse effects on the evaluation.
>
> ### Bibliography
>
> [1] Zimmermann et al. (2021): “Contrastive Learning Inverts the Data Generating Process”
>
> [2] Von Kügelgen et al. (2021): “Self-Supervised Learning with Data Augmentations Provably Isolates Content from Style”
>
> [3] Rusak et al. (2024): “InfoNCE: Identifying the Gap Between Theory and Practice”
>
> [4] Simon et al. (2020): “On the Stepwise Nature of Self-Supervised Learning”
>
> [5] Jing et al. (2021): "Understanding Dimensional Collapse in Contrastive Self-supervised Learning"

---

> > ### Author Response · Authors · 2024-11-22
> >
> > ### W2: Synthetic misspecifications
> >
> > We agree with the reviewer that our experiments showcasing the (lack of) robustness to assumption violations should be more extensive.
> >
> > We appreciate the suggestion to explore the Mixture of Gaussians case (or, similarly, the Mixture of vMFs). In this setup, each cluster is represented by a set of fixed, proximal vectors, and samples are drawn from one of these vectors according to a Gaussian distribution (or a vMF, provided the proximal vectors lie on the hypersphere). The issue with this setup is that it is also identifiable up to a linear transformation. **The Mixture of vMFs case is a direct application of the results already proven in our paper**, with the fixed proximal vectors serving as the new cluster vectors. While the Gaussian case (when samples may fall of the hypersphere) is not directly covered by our theorem, it follows a similar reasoning and has an analogous proof to ours.
> >
> > Instead, **we explore the amount of violation caused by distributions of varying scales and shapes**. We do this by utilising a _generalised Normal distribution_ (where the shape parameter $\beta=1$ corresponds to a Laplace and $\beta=2$ to a Normal distribution) and, furthermore, we vary the concentration parameter alpha (that corresponds to what we previously also called concentration parameter in the case of the vMF conditional).
> > In Appx. D and Fig. 4, the left plot reports the recovery of latent $z$, the middle the recovery of cluster vectors $v_c$, and the right the relationship between the concentration parameter and the average norm of representations (in a 1D case). These scores are the average across 5 seeds.
> >
> > In Appx. D, Fig. 5, we also show that **the fat tails of the Laplace distribution make it significantly deviate from the vMF case**, leading to a much lower identifiability scores. To show this this, we use a _truncated Laplace_ conditional, where the truncation parameter restricts the distribution’s support to the symmetrical interval [-Truncation;Truncation]. As Fig. 5 Right demonstrates, a higher truncation parameter (i.e., a distribution with **fatter tails**) **corresponds to lower $R^2$ scores**, this relationship holds over multiple concentration parameters and seeds.
> >
> > From these figures, we conclude:
> > - [Fig. 4 Left and Middle] we see that the **distribution's shape and concentration plays a critical role**. Moving towards a Laplace distribution leads to more performance degradation, which is due to its fatter tails (shown in Fig. 5 Right). This difference is even more pronounced for high concentration values.
> > - [Fig. 4 Right] We show that there is a **non-trivial relationship between the norm of the representation and the concentration as a function of the shape parameter $\beta$**. For a given concentration value $\alpha$, decreasing beta (i.e., moving towards a Laplace distribution) leads to a bigger average norm.
> >
> > ### W3: Weak ImageNet-X results
> >
> > We re-evaluated our results on ImageNet-X and now provide statistical tests to have a better overview on the performance of trained models in recovering (proxy) latent features. As our **revised Fig. 3** suggests, **almost all features are recovered better than chance performance** (except “brighter” - second subplot in the second row) both for trained ViTs and ResNet50. More importantly, as **there are no true latents available for ImageNet-X, the ones we evaluate against are only binary proxies defined by human evaluators**. We hypothesize that the proxy nature and especially the binary latents are the cause of degraded performance
> >
> >
> > ### Q1: The impact of repeated elements
> >
> > The questions is interesting and relevant. We distinguish the theoretical and more practical cases.
> >
> > In our theoretical setup, the data generating process (DGP) is defined in a way that the **occurrence of repeated elements is bound to happen**. Namely, the vMF conditional distribution has full support over the hypersphere and, hence, any sample could have resulted from any instance. **Nonetheless, the prediction of the theorem holds**: the inferred parameter vectors $w_i$’s pick up the values of the cluster vectors that correspond to the instance label $i$. The latents are reconstructed up to linear transformations and the score is unchanged.
> >
> > The phenomenon may seem more perplexing **in practical, finite dataset cases. Here, seemingly the same sample is mapped to two different indices**. Yet, this _does not change the learned representation or the corresponding parameter vector $w$_. The two parameter vectors $w_i$ and $w_j$ will, in fact, match. **What changes is the value of the instance discrimination objective (DIET)**, as the model is now uncertain in assigning the index. The logit values of index $i$ and $j$ will match. However, again, this does not affect the learned representation.

---

> > ### Comment · Reviewer_vcau · 2024-11-22
> >
> > Thank you for the clear explanation! I am happy to keep my recommendation to accept your paper.

---

### Official Review · Reviewer_8oBq · 2024-11-03

**Soundness:** 3
**Presentation:** 3
**Contribution:** 3
**Rating:** 8
**Confidence:** 4

**Summary:**

This paper provides a theoretical framework to further advance the linear identifiability of self supervised learning, and further to extend to general supervised classification tasks where cross-entropy loss is used.  The theoretical framework is constructed on a clever data generating process on the clusters of von-Mises-Fisher distribution.  Experimental results are included to empirically demonstrate the validity of the theoretical analysis. Empirical experiments on ImageNet-X dataset are also included.

**Strengths:**

The framework poses significant progress towards better theoretical understanding on machine learning tasks, particularly classification tasks. If the theory holds, it may have a substantial influence on theories of adjacent research areas as well as many applications. Although this is a theoretical paper, the author(s) have made efforts to make the writing approachable, e.g., drawing from practical observations such as interpretable embeddings across input pairs from Mikolov 2013, and linear superposisiton in Elhage 2022.

**Weaknesses:**

My main concern is on how the author(s) conduct empirical validation. Table 2 and 3 appears to be the main empirical validation of the proposed hypothesis. The author(s) have devised a clever method for validation. However the statstics in Table 2 and 3 only shows that when the Assumptions 1 is true, the DIET method can successfully recover the cluster vectors with high probability. Another set of the experiments should be conducted to demonstrate that the hypothesis can be "falsified" when the assumptions are no longe true. This can be easily done with the current experiment setup:
1. After the data in each cluster are generated, choose 2 clusters, and random swap a percentage of the samples between the two, report the statistics of the recovery of all clusters. If the hypothesis holds, the outcome should be clear
2. Increase the percentage of the samples randomly swapped, the trend should be clear.
3. Repeat 1 and 2, but with more clusters involved. The trend represented by the statistics should also be quite clear

**Questions:**

Reiterate the need of additional experimental results, please consider doing the following:

1. After generating data in each cluster, randomly swap a percentage of samples between two clusters and report recovery statistics for all clusters.
2. Increase the percentage of randomly swapped samples and observe the trend.
3. Repeat this process with more clusters involved.

This would provide a more comprehensive evaluation by demonstrating both when the hypothesis holds and when it breaks down under violations of the clustering assumptions.  I'd like to see the experimental results discussed in the section above. In my view, it should be easy to conduct and will strengthen the paper.

The details of the experiments on DisLib and ImageNet-X should be included in appendix and/or a open github repo. DisLib (or at least a subset of it) has been widely used in many disengagement papers. The code or details explanation of the experiments should be provided for other researchers to compare, it is understandable that this experiment is not the main focus of this manuscript, including:

1. Hyperparameters used
2. Data preprocessing steps
3. Evaluation metrics
4. Any modifications made to standard implementations

---

> ### Author Response · Authors · 2024-11-22
>
> We thank Reviewer 8oBq for praising our submission’s **“significant progress towards better theoretical understanding on… classification tasks”**, its potentially **“substantial influence on theories of adjacent research areas as well as many applications”**, and our **“efforts to make the writing approachable”**
> We are also grateful for the many constructive suggestions, for which we provide answers below.
>
> ### Label Swapping Experiments
> We appreciate the reviewer’s experimental suggestion, which provides an approach to evaluate the robustness of our method under increasing violations of the assumptions. In the revised manuscript, we have incorporated a **new set of experiments to assess robustness to label noise**, now detailed in **Appx. D and Fig. 6**. Appx. D also contains the following discussion.
>
> For the DIET case (Figure 6 Left), the experiment was conducted as follows: after generating a sample, the instance label was perturbed with a probability equal to the label noise ratio (x-axis in the figures). Perturbed samples received a new label drawn uniformly from the set of all instance labels. For the supervised case (Figure 6 Right), the cluster label was perturbed instead. In both cases, the y-axis represents the identifiability score ($R^2$ score).
>
> This experiment addresses points 1 and 2 of the Reviewer in a general manner, as the mislabeling involves all clusters rather than just two. Furthermore, it is a concrete interpretation of suggestion 3 (“swapping labels from multiple clusters”). We believe this setup reflects realistic scenarios, where label noise is equally likely to affect any data point, as commonly assumed in the literature [1,2].
> Despite this increasingly challenging setup, the **results demonstrate remarkable robustness to label noise**. Up to an 80% label noise ratio, latent recovery shows minimal degradation, and the cluster recovery performance of DIET remains perfect. We attribute this robustness to the symmetry of label noise across all instances and labels. While increased uncertainty does reduce the accuracy of individual label predictions, the optimal logit values predicted by the encoder shouldn’t change under symmetrical label noise. The stability of deep learning models to label noise has also been shown by [3].
>
> ### Experimental details on DisLib and ImangetNet-X
> Thank you for pointing out the missing experimental details. We **added the missing hyperparameters, the references to the implementation we used, the evaluation metrics and our evaluation protocol in Appendix C**. We also provide **our code as supplementary material** and will release the repository in GitHub upon acceptance.
>
>
> ### Bibliography
> ​​[1] Nettleton et al. (2010): "A study of the effect of different types of noise on the precision of supervised learning techniques." \
> [2] Frénay & Verleysen (2014): "Classification in the presence of label noise: a survey." \
> [3] Rolnick et al. (2017): "Deep learning is robust to massive label noise."

---

> > ### Comment · Reviewer_8oBq · 2024-11-25
> > **Intriguing experimental results**
> >
> > thanks for the updates. I was expecting more gradual transition as the percentage of the swapping increases. There are a few typos in the caption of Fig. 6. Please correct.
> >
> > I am revising my rating based on the responses and revision.

---

> > > ### Author Response · Authors · 2024-11-25
> > >
> > > Thank you very much! We will fix the typos in the caption.

---

> ### Author Response · Authors · 2024-11-25
>
> Dear Reviewer 8oBq,
>
> Thank you again for your constructive feedback! We have updated our submission with your proposed experiments, described the requested experimental details, and attached our code as supplementary material. Please let us know if you have any further questions.

---

### Meta-Review · Area_Chair_98XB · 2024-12-25

**Metareview:**

The paper presents identifiability results of  latent representations learned with cross entropy loss under assumption on the data  and labels generation distribution (a clustering model with von Mises-Fisher samples around the cluster centers). Under these assumptions the latent can be recovered from data up to a linear transformation. Authors show the validity of their results on synthetic data and on the  the DisLib, a disentanglement benchmark with known factors of variations. Finally on the more challenge imagenet-X , positive correlations of the inferred latents and the true one are shown but with weaker correlations. Accept.

**Additional Comments On Reviewer Discussion:**

Reviewers agreed that the contribution of the paper is interesting and insightful  to the ICLR contribution and discussed the paper with the authors

---

### Decision · Program_Chairs · 2025-01-22

Accept (Oral)